# VGPA: Deep View-Graph Pose Averaging for Structure-from-Motion

## Abstract

Camera pose estimation is a key step in 3D reconstruction and view-synthesis pipelines. We present a deep, global Structure-from-Motion framework based on learned view-graph aggregation. Our method employs a permutation-equivariant, edge-conditioned graph neural network that takes noisy pairwise relative poses as input and outputs globally consistent camera extrinsics. The network is trained without ground-truth supervision, relying solely on a relative-pose consistency objective. This is followed by 3D point triangulation and robust bundle adjustment. A fast view re-integration step increases camera coverage by reintroducing discarded images. Our approach is efficient, scalable to more than a thousand images, and robust to graph density. We evaluate our method on MegaDepth, 1DSfM, Strecha, and BlendedMVS. These experiments demonstrate that our method achieves superior rotation and translation accuracy compared to deep track-centric methods while registering more images across many scenes, and competitive results compared to state-of-the-art classical pipelines, while being much faster.

## 1 Introduction

Camera pose recovery is an essential part of 3D scene reconstruction and view synthesis applications. Many common Multiview Stereo (MVS) (Seitz et al., 2006; Yao et al., 2018) and view synthesis methods, including Neural Radiance Fields (NeRF) (Mildenhall et al., 2021) and Gaussian Splatting (GS) (Kerbl et al., 2023) rely on accurate camera poses computed in preprocessing. View synthesis methods, in particular, have gained much popularity in recent years, as they can produce novel, realistically looking images and walkthroughs for complex scenes.

Multiview Structure-from-Motion (SfM) techniques provide reliable tools for camera pose recovery. Sequential pipelines, e.g., COLMAP (Schönberger & Frahm, 2016), solve for one camera at a time, enriching the recovered set of camera poses and 3D points by processing image by image. These, generally highly accurate techniques, are relatively slow when applied to large collections of images, and their performance depends on the order in which the images are processed. Projective factorization techniques (Sturm & Triggs, 1996) simultaneously solve for all cameras and point tracks. These methods, however, attempt to factor large tensors that include all the track points.

In the past decade, *global methods* emerged as an alternative to sequential and factorization methods. Global methods use a technique called *motion averaging*; given pairwise relative camera motion measurements, they seek to recover the location and orientation (and possibly also the intrinsic parameters) of cameras in a global coordinate system. Typically, this is done by solving separately for rotations and translations (Moulon et al., 2016; Sweeney et al., 2015), while some recent works developed techniques for directly averaging essential and fundamental matrices (Kasten et al., 2019a;b). Global methods can be more efficient than both sequential and factorization-based techniques, as they only solve for pose and therefore do not need to access and manipulate point tracks, except in the final bundle adjustment (BA) step.

In this paper, we reexamine the use of global SfM through the lens of *learned view-graph aggregation*. Specifically, we propose an efficient permutation-equivariant, edge-conditioned graph neural network (GNN) that takes as input noisy estimates of pairwise relative camera poses associated with the edges of a view graph, and outputs globally consistent camera extrinsics. The network is trained *without ground-truth supervision* using only a relative-pose consistency objective. Unlike existing

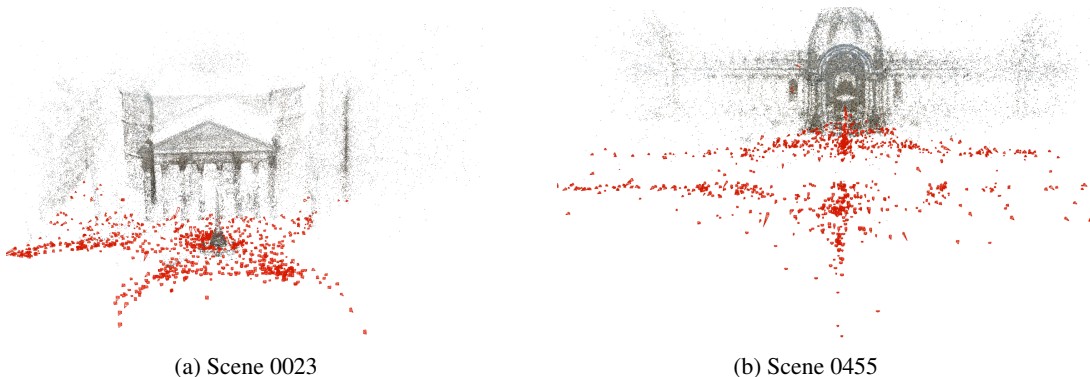

(a) Scene 0023                     (b) Scene 0455

Figure 1: 3D reconstructions and recovered camera parameters produced by VGPA on two large scenes ($N_c > 1000$ images). VGPA registers almost all images and scales to thousand-image collections, in contrast to existing image-based deep methods (e.g., VGGT, VGGSfM).

deep-based approaches to SfM (Khatib et al., 2025; Moran et al., 2021; Brynte et al., 2023), our pose regression network does not use point tracks; it does not predict 3D points and does not rely on a reprojection loss. At test time, we use our network to predict global camera poses. Then, we improve our camera pose predictions by triangulating point tracks and applying robust BA. Finally, an optional and efficient *view reintegration* step is applied to recover cameras that were discarded in the process by the network, increasing camera coverage.

Our approach is efficient and achieves high accuracy. It copes well with large-scale inputs, including ones with more than a thousand images. Moreover, our method is agnostic to the density of the graph. We obtain comparable performance when constructing the view graph using top-$k$ neighbors retrieved with NetVLAD (Arandjelovic et al., 2016), instead of exhaustive pairwise matching, despite the large difference in edge density. We note that in the uncalibrated setting, we optimize jointly for the intrinsics and extrinsics parameters during BA.

We perform an extensive experimental evaluation on challenging datasets, including MegaDepth and 1DSfM. These experiments demonstrate that our learned pose averaging achieves lower camera position and orientation errors than existing deep track-centric methods while registering more images on many scenes (Khatib et al., 2025; Moran et al., 2021; Brynte et al., 2023), and is competitive with strong classical pipelines. Similar results are obtained on smaller calibrated benchmarks for which ground truth measurements are available (Strecha and BlendedMVS) and on scenes containing challenging cyclic trajectories, where reprojection-centric methods such as (Khatib et al., 2025; Moran et al., 2021; Brynte et al., 2023) often struggle.

Below we summarize our contributions.

1. We present **VGPA**: an efficient, permutation-equivariant GNN for *view-graph pose averaging* that predicts global camera extrinsics from noisy pairwise estimates.

2. Our method achieves highly accurate camera pose and structure recovery, comparable to state-of-the-art classical methods while being much faster, and it largely outperforms recent deep-based methods on large-scale scenes.

3. We train VGPA in a **self-supervised manner by enforcing relative-pose consistency only**; structure is recovered via triangulation followed by robust BA.

4. We show **robustness to view-graph density**, achieving similar accuracy with both exhaustive pairwise matching and sparse top-$k$ NetVLAD graphs, despite large differences in edge count.

5. **Handles unknown intrinsics:** VGPA remains accurate when intrinsics are coarsely initialized and optimized only during BA.

6. We introduce **a lightweight technical view re-integration step** that optionally improves camera coverage with minimal runtime overhead.

## 2    RELATED WORK

A popular classical method for Structure-from-Motion (SfM) uses an incremental algorithm in which images are processed one at a time, gradually extending the recovered set of camera poses and 3D structure. (Agarwal et al., 2011; Schönberger & Frahm, 2016; Snavely et al., 2006; Wu, 2013). While these methods achieve highly accurate reconstruction, they are inefficient when applied to large image collections, and their results depend on the order in which images are processed.

A second approach uses projective factorization to solve simultaneously for camera pose and 3D structure on all input images (Sturm & Triggs, 1996; Dai et al., 2010; Lin et al., 2017). This method uses the observation that point track matrices are rank 4 when the points are scaled properly. Classical algorithms based on SVD factorization, however, are restricted to uncalibrated settings and do not handle missing data or outliers. Inspired by these techniques, several recent works train equivariant network architectures to jointly estimate camera poses and 3D structure from point tracks (Moran et al., 2021; Brynte et al., 2023; Chen et al., 2024; Khatib et al., 2025). These methods use either set-of-sets or graph transformer network architectures and are trained with either supervised or unsupervised data. An inlier/outlier classifier is incorporated for improved robustness (Khatib et al., 2025). Accurate pose recovery results were achieved with this method. However, it tends to over-prune valid inliers, leading to occasional registration failures and reduced image coverage.

Our method follows a third approach, commonly referred to as a *global approach*. Global methods handle all images simultaneously by applying manifold averaging to ensure the consistency of pairwise pose relations (rotations and translations) inferred from the essential matrices. Existing methods commonly solve first for camera orientations, and next for location and scales (Martinec & Pajdla, 2007; Özyeşil et al., 2017; Sweeney et al., 2015; Moulon et al., 2016). Kasten et al. (2019a;b) introduced an averaging method for averaging essential and fundamental matrices, solving for all of these parameters in a single optimization. With the exception of (Pan et al., 2024), these methods require a separate step of 3D point triangulation. Theia (Sweeney et al., 2015) and the recent GLOMAP (Pan et al., 2024), in particular, were shown to yield accurate recovery.

Several recent works train networks to solve rotation averaging on the view graph. NeurORA (Purkait et al., 2020) learns to denoise pairwise relative rotations and aggregates them to recover global orientations, while (Li & Ling, 2021) applies message passing on pose graphs to iteratively update node rotations. These methods only address rotation averaging; they are trained on supervised data and tested in limited settings that do not include cross-dataset generalization. In contrast, our method is trained with unsupervised data and recovers the full camera extrinsics.

Recent learnable SfM methods such as VGGSfM (Wang et al., 2023a), DUST3R (Wang et al., 2023b), and MAST3R (Leroy et al., 2024) are restricted to processing only a small number of input images, whereas Ace-Zero (Brachmann et al., 2024) and FlowMap (Smith et al., 2024) are tailored for video sequences under constant illumination. More recently, VGGT (Wang et al., 2025) introduced an end-to-end transformer that jointly predicts camera poses, dense 3D structure, and point tracks. Although promising, VGGT requires substantial supervised training and is currently restricted to images on the order of $\sim 200$. Fast3R (Yang et al., 2025) scales to larger collections but typically attains lower accuracy than VGGT at comparable settings.

In this paper, we introduce a learned view-graph pose averaging module implemented with a permutation-equivariant graph neural network. Trained without ground-truth supervision, our method achieves competitive accuracies at lower runtime than strong global SfM baselines and surpasses prior deep factorization approaches in both accuracy and camera coverage. It remains robust to heavy outlier contamination in realistic point track data.

## 3    METHOD

Given a collection of $m$ images of a stationary scene, we assume, as in standard SfM pipelines, that in preprocessing we extract (1) essential matrices and (2) a collection of point tracks, which will form the input to our pipeline. Our objective is to recover the camera matrices for all the given images and a triangulated 3D location for each track. Below, we describe each step in our method.

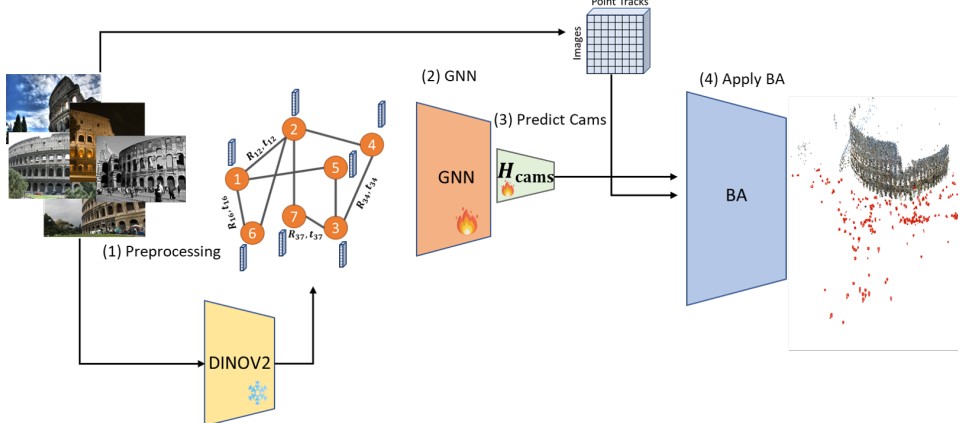

Figure 2: **Method overview.** (1) *Preprocessing:* estimate pairwise relative poses from essential matrices and build the view graph; extract point tracks and frozen DINOv2 image descriptors. (2) *GNN:* a permutation-equivariant, edge-conditioned GNN aggregates the view graph to produce camera embeddings. (3) *Predict cams:* a small head $H_{\text{cams}}$ regresses global extrinsics $(R_i, \mathbf{t}_i)$ from the embeddings. (4) *Triangulation + BA:* using the predicted cameras and the point tracks, we triangulate 3D points and run robust bundle adjustment.

### 3.1 PREPROCESSING

Denote our input images by $I_1, ..., I_m$. Following standard SfM pipelines, we begin by detecting and matching features across the images using standard algorithms such as SIFT or SuperPoint (DeTone et al., 2018; Lowe, 2004). We next apply RANSAC (Bolles & Fischler, 1981) and obtain a partial collection of pairwise essential matrices $\{E_{ij}\}_{i,j\in[m]}$, denoted by $\mathcal{E}$. Each essential matrix encodes the relative rotation $R_{ij}$ and translation $\mathbf{t}_{ij}$ between camera $P_i$ and $P_j$. We extract the rotation and translation by decomposing the essential matrix, while enforcing positive depth. Note that $\mathbf{t}_{ij}$ is determined at this point only up to scale. These pairwise rotation and translation measurements serve as input to our pose averaging module.

A second outcome of the procedure above comprises pairs of matched feature points across images. We next use heuristics (as in, e.g., (Schönberger & Frahm, 2016)) to join such pairs to form longer tracks. Each track is a set $T_k = \{\mathbf{x}_{i_1,k}, \mathbf{x}_{i_2,k}, ...\}$ with $i_1, i_2, ... \in [m]$, and we assume that $T_k$ contains the projected locations of a single 3D scene point, denoted $\mathbf{X}_k$, onto $I_{i_1}, I_{i_2}, ....$. These tracks are generally contaminated by small displacement errors (noisy measurements) and outliers. We will use this collection of point tracks at a later stage to triangulate the 3D structure using predicted absolute camera poses.

### 3.2 NETWORK ARCHITECTURE

Our network applies *pose averaging* to the view graph. As is shown in Fig. 2, it comprises two modules: (i) a permutation-equivariant, edge-conditioned GNN that aggregates pairwise relative poses into camera embeddings; and (ii) a regression head that predicts global camera parameters from these embeddings.

**Pose-averaging GNN.** We build a viewing graph $\mathcal{G} = (\mathcal{V}, \mathcal{E})$ whose nodes index the $m$ images and whose edges carry relative-pose measurements. For each edge $(i, j) \in \mathcal{E}$ we define

$$\mathbf{e}_{ij}^{(0)} = \phi_e\big([\log R_{ij}^{\text{RANSAC}}, \mathbf{t}_{ij}^{\text{RANSAC}}]\big),$$

where $\log : SO(3) \to \mathfrak{so}(3)$ is the matrix logarithm and $\mathbf{t}_{ij}^{\text{RANSAC}} \in \mathbb{S}^2$ is the *unit normed*-translation direction recovered from the essential matrix. To inject image-level context, each node $v_i$ is initialized with the DINOv2 [CLS] token $\mathbf{h}_i^{(0)}$ computed from image $I_i$.

We apply edge-conditioned message passing with degree-normalized mean aggregation:

$$\tilde{\mathbf{m}}_i^{(\ell)} = \sum_{j \in \mathcal{N}(i)} \phi_m(\mathbf{h}_i^{(\ell)}, \mathbf{h}_j^{(\ell)}, \mathbf{e}_{ij}^{(\ell)}),$$

$$\mathbf{m}_i^{(\ell)} = \frac{1}{|\mathcal{N}(i)|} \tilde{\mathbf{m}}_i^{(\ell)},$$

$$\mathbf{h}_i^{(\ell+1)} = \mathrm{LN}\Big(\mathbf{h}_i^{(\ell)} + \mathrm{Drop}\big(\psi\big(\mathrm{LN}(\mathbf{h}_i^{(\ell)}), \mathbf{m}_i^{(\ell)}\big)\big)\Big),$$

for $\ell = 0, \ldots, L-1$, where $\phi_m$ and $\psi$ are MLPs, LN denotes Layer Normalization (Ba et al., 2016), and Drop denotes Dropout (Srivastava et al., 2014). The network is equivariant to node relabelings (permutations) of $\mathcal{G}$. The final node embeddings are $z_i = \mathbf{h}_i^{(L)}$.

**Pose regression head.** The pose regression head obtains as input the per-camera embeddings $\mathbf{z}_i$ produced by the pose-averaging GNN. A 3-layer MLP head $H_{\mathrm{cams}}$ maps these embeddings to camera parameters,

$$(\mathbf{t}_i, \mathbf{q}_i) = H_{\mathrm{cams}}(\mathbf{z}_i), \qquad \tilde{\mathbf{q}}_i \leftarrow \mathbf{q}_i / \|\mathbf{q}_i\|,$$

where $\mathbf{t}_i \in \mathbb{R}^3$ and $\tilde{\mathbf{q}}_i \in \mathbb{H}$ is a unit quaternion.

### 3.3 OUTPUT AND LOSS

Our network predicts the $m$ internally calibrated cameras $P_1, \ldots, P_m$. Each camera is parameterized as $P_i = [R_i \mid \mathbf{t}_i]$ with $R_i \in SO(3)$ and $\mathbf{t}_i \in \mathbb{R}^3$; the camera center is $-R_i^\top \mathbf{t}_i$.

Training is unsupervised and seeks cameras $P_1, \ldots, P_m$ that best agree with the pairwise relative-pose estimates. We therefore minimize a relative-pose consistency objective. Specifically, we use

$$\mathcal{L}_{\mathrm{RelPose}} = \frac{1}{|\mathcal{E}|} \sum_{(i,j) \in \mathcal{E}} d_R\big(\hat{R}_{ij}, R_{ij}^{\mathrm{RANSAC}}\big) + \frac{1}{|\mathcal{E}|} \sum_{(i,j) \in \mathcal{E}} d_t\big(\hat{\mathbf{t}}_{ij}, \mathbf{t}_{ij}^{\mathrm{RANSAC}}\big), \tag{1}$$

where $\hat{R}_{ij}$ and $\hat{\mathbf{t}}_{ij}$ denote the relative rotation and translation estimated from the output cameras $P_i$ and $P_j$ using

$$\hat{R}_{ij} = R_j^T R_i, \qquad \hat{\mathbf{t}}_{ij} = R_j^T(\mathbf{t}_i - \mathbf{t}_j), \tag{2}$$

$R_{ij}^{\mathrm{RANSAC}}$ and $\mathbf{t}_{ij}^{\mathrm{RANSAC}}$ are the corresponding rotation and translation obtained with RANSAC in preprocessing, $d_R(R_1, R_2) = \arccos\left(\frac{\mathrm{trace}(R_1^\top R_2) - 1}{2}\right)$ is the geodesic rotation error, and $d_t(\mathbf{a}, \mathbf{b}) = \arccos\langle \mathbf{a}, \mathbf{b} \rangle$ measures directional disagreement.

**Training protocol.** We iterate over all training scenes in each epoch. For each scene, we sample uniformly $s \in [0.1, 0.2]$ of the images (without replacement) to form a subgraph. Models are selected by early stopping on a held-out validation set; we report the checkpoint with the lowest validation error. Additional details appear in the Appendix.

**Inference.** On an *unseen* scene, the model predicts all camera poses in a single forward pass. We then fine-tune on the target scene with the unsupervised objective (no ground-truth labels) for $T_{\mathrm{ft}} = 200$ steps. Next, we triangulate using DLT (Hartley & Zisserman, 2003) to recover 3D point positions from the estimated camera poses and point tracks, and finally perform a robust bundle adjustment initialized with the camera poses predicted by the network and the triangulated points.

## 4 EXPERIMENTS

### 4.1 DATASETS

We train our network on scenes from the MegaDepth dataset (Li & Snavely, 2018) and then test it on a diverse range of real-world scenes that include novel scenes from the MegaDepth dataset as well as cross-dataset generalization tests on the 1DSfM dataset (Wilson & Snavely, 2014), Strecha (Strecha et al., 2008), and BlendedMVS (Yao et al., 2020). We refer the reader to the supplementary material for hyperparameters and further technical details.

**MegaDepth (Li & Snavely, 2018).** The MegaDepth dataset includes 196 different outdoor landmark scenes curated from the internet. We followed the train/test split as in (Khatib et al., 2025), including subsampling of scenes with more than 1000 images. In Table 1, above the middle rule are scenes with fewer than 1000 images, while the scenes below the rule are subsampled.

**1DSFM (Wilson & Snavely, 2014).** 1DSFM is a collection of diverse urban scenes reconstructed from community photo collections. We use this dataset to test our method (trained on the MegaDepth dataset) in cross-dataset generalization experiments, demonstrating large-scale reconstructions in realistic settings.

**Strecha (Strecha et al., 2008).** The Strecha dataset consists of five small outdoor scenes ($\leq 30$ images) and includes ground-truth data acquired with a LIDAR system. We test our method on four of these five scenes.

**BlendedMVS (Yao et al., 2020).** The BlendedMVS dataset includes synthetic scenes with textured meshes rendered and blended to produce color images and depth maps, providing ground truth camera poses.

**Ground truth camera poses.** Many challenging datasets, including MegaDepth and 1DSFM, lack ground truth measurement, and, therefore, as is common in the field, we use camera poses computed with COLMAP Schönberger & Frahm (2016), a state-of-the-art incremental Structure from Motion (SfM) method, to generate "ground truth" camera poses. COLMAP is widely used for this purpose (see Jiang et al. (2013); Wilson & Snavely (2014); Cui & Tan (2015); Ozyesil & Singer (2015); Brynte et al. (2023); Khatib et al. (2025); Zhang et al. (2024)) due to its accurate and robust performance. To evaluate our method with real ground truth, we additionally show results on the smaller datasets Strecha (Strecha et al., 2008) and BlendedMVS (Yao et al., 2020).

## 4.2 BASELINES

With the exception of VGGT (Wang et al., 2025), the settings and results for all baselines below were taken from (Khatib et al., 2025).

**RESfM (Khatib et al., 2025).** RESfM is a robust deep equivariant SfM model that operates on a point-track tensor using a sets-of-sets permutation-equivariant architecture. It augments prior equivariant factorization by adding a multiview inlier/outlier classifier integrated into the same equivariant backbone and concludes with a robust bundle-adjustment stage.

**VGGSfM (Wang et al., 2024)** is a differentiable, trainable SfM pipeline.

**MASt3R (Leroy et al., 2024).** An SfM pipeline that utilizes a global alignment procedure to merge pairwise pointmap predictions.

**Theia (Sweeney et al., 2015).** A global SfM pipeline that applies rotation averaging, followed by translation averaging, and finally 3D point triangulation.

**GLOMAP (Pan et al., 2024).** A global SfM pipeline that first applies rotation averaging, followed by an integrated step of translation averaging and point triangulation.

**VGGT (Wang et al., 2025).** VGGT is a feed-forward, end-to-end multi-view transformer network that jointly predicts cameras, depth, point maps, and tracks for up to about 200 views. It uses alternating inter-frame/global attention and is additionally refined with BA.

## 4.3 METRICS AND EVALUATION

To evaluate our results, we first align the predicted scenes to the ground truth by applying a per-scene 3D similarity transformation. We then compare our camera orientation predictions with the ground truth ones using angular differences in degrees. We measure differences between our predicted and ground truth camera locations using the $l_2$ distance. For a fair comparison, both our method and all the baseline methods (except VGGSfM, VGGT and MAST3R, which are applied directly to the input images) were run with the same set of point tracks. For all methods, we apply a final post-processing step of robust bundle adjustment.

Table 1: **MegaDepth experiment.** For each scene, we show the number of input images (denoted $N_c$) and the fraction of outliers. For each model, we show the number of images used for reconstruction (denoted $N_r$) and mean values of the rotation (in degrees) and translation errors. (Above the middle rule are Group 1 scenes with $< 1000$ images; below are Group 2 scenes with $> 1000$ images, subsampled to 300 for testing.) Winning results are marked in bold and underlined. Yellow represents the best result among the deep-based algorithms and green among the classical algorithms.

| Scene | $N_c$ | Outliers% | **Ours** | | | RESfM | | | Theia | | | GLOMAP | | |
| --- | --- | --- | --- | --- | --- | --- | --- | --- | --- | --- | --- | --- | --- | --- |
| | | | $N_r$ | Rot | Trans | $N_r$ | Rot | Trans | $N_r$ | Rot | Trans | $N_r$ | Rot | Trans |
| 0238 | 522 | 44.6% | 488 | 4.50 | 0.686 | 283 | 2.61 | **0.325** | 506 | 1.21 | 0.334 | 499 | **0.74** | 0.349 |
| 0060 | 528 | 41.6% | 518 | **0.07** | **0.014** | 503 | 0.29 | 0.029 | **525** | 0.85 | 0.124 | 522 | 0.11 | 0.048 |
| 0197 | 870 | 40.7% | 641 | 1.28 | 0.271 | 667 | 4.22 | 0.333 | **855** | 1.16 | 0.227 | 814 | **0.43** | **0.129** |
| 0094 | 763 | 40.1% | 663 | **0.66** | **0.101** | 537 | 3.77 | 0.750 | **742** | 0.75 | 0.160 | 717 | 0.88 | 3.907 |
| 0265 | 571 | 38.8% | 345 | 2.93 | 0.998 | 346 | **1.25** | **0.389** | 554 | 5.83 | 2.216 | **558** | 7.46 | 2.839 |
| 0083 | 635 | 31.3% | 614 | **0.06** | **0.005** | 596 | 0.64 | 0.058 | **632** | 0.37 | 0.372 | 614 | 0.08 | 0.016 |
| 0076 | 558 | 30.5% | 543 | **0.09** | **0.016** | 524 | 0.37 | 0.094 | **549** | 0.78 | 0.120 | 541 | 0.17 | 0.042 |
| 0185 | 368 | 30.0% | 358 | 0.10 | 0.022 | 350 | **0.06** | **0.010** | **365** | 0.41 | 0.094 | **365** | 0.16 | 0.051 |
| 0048 | 512 | 24.2% | 500 | 0.29 | **0.026** | 474 | 4.69 | 0.178 | **507** | 0.41 | 0.105 | 506 | **0.15** | 0.224 |
| 0024 | 356 | 23.0% | 313 | 3.38 | 0.772 | 309 | 2.03 | 0.398 | **355** | 0.56 | 0.219 | 339 | **0.15** | **0.104** |
| 0223 | 214 | 17.0% | 208 | 2.75 | **0.195** | 204 | 3.76 | 0.510 | 212 | 3.34 | 0.519 | **214** | 1.75 | 0.275 |
| 5016 | 28 | 16.9% | **28** | **0.08** | **0.015** | 28 | 0.12 | 0.016 | 28 | 0.10 | 0.061 | **28** | **0.08** | 0.046 |
| 0046 | 440 | 14.6% | 439 | 0.54 | 0.071 | 399 | 0.95 | **0.043** | 434 | 0.25 | 0.112 | **440** | **0.03** | **0.007** |
| 1001 | 285 | 43.9% | 265 | 1.89 | 3.840 | 251 | **1.70** | **0.661** | 276 | 7.97 | 4.014 | **281** | 4.56 | 3.817 |
| 0231 | 296 | 42.2% | 261 | **0.24** | **0.030** | 246 | 0.84 | 0.065 | **286** | 1.37 | 0.322 | 279 | 0.73 | 0.134 |
| 0411 | 299 | 29.9% | 270 | **0.12** | **0.018** | 273 | 0.13 | 0.020 | **293** | 0.39 | 0.196 | 269 | 0.19 | 0.148 |
| 0377 | 295 | 27.5% | 232 | 0.30 | 0.035 | 210 | **0.29** | **0.018** | **269** | 1.13 | 0.205 | 268 | 0.65 | 0.237 |
| 0102 | 299 | 25.8% | **297** | 0.18 | **0.031** | 284 | 0.28 | 0.059 | 294 | 2.31 | 0.698 | 293 | **0.15** | 0.101 |
| 0147 | 298 | 24.6% | 282 | 1.99 | 0.153 | 207 | 4.62 | 0.325 | 284 | 6.36 | 0.934 | **290** | 6.75 | 3.542 |
| 0148 | 287 | 24.6% | 211 | 0.93 | 0.037 | 197 | **0.60** | **0.035** | 275 | 13.98 | 1.558 | **283** | 22.73 | 2.646 |
| 0446 | 298 | 22.1% | 292 | **0.22** | **0.019** | 288 | 0.72 | 0.046 | 289 | 1.23 | 0.391 | **296** | **0.20** | 0.071 |
| 0022 | 297 | 21.2% | 277 | 0.29 | 0.044 | 274 | 0.29 | **0.039** | **296** | 0.58 | 0.160 | 281 | **0.22** | 0.087 |
| 0327 | 298 | 21.0% | **291** | **0.12** | **0.014** | 271 | 0.26 | 0.090 | 288 | 1.27 | 0.360 | 290 | 15.54 | 2.035 |
| 0015 | 284 | 20.6% | 243 | 0.52 | **0.058** | 215 | 1.04 | 0.167 | 244 | 2.21 | 0.389 | **274** | **0.28** | 0.095 |
| 0455 | 298 | 19.8% | 290 | 0.39 | 0.078 | **293** | 0.68 | 0.105 | 294 | 0.77 | 0.159 | **298** | **0.35** | **0.064** |
| 0496 | 297 | 19.2% | 279 | 0.37 | **0.033** | 281 | **0.35** | 0.055 | 285 | 1.40 | 0.550 | **291** | 0.44 | 0.303 |
| 1589 | 299 | 17.4% | 296 | 0.11 | **0.010** | 290 | 0.14 | 0.019 | 288 | 0.82 | 0.193 | **299** | **0.07** | 0.041 |
| 0012 | 299 | 16.3% | **295** | 0.63 | 0.071 | 287 | **0.40** | **0.027** | 129 | 1.04 | 0.318 | **295** | 0.51 | 0.121 |
| 0019 | 299 | 15.4% | 291 | 0.37 | 0.020 | 250 | **0.06** | **0.008** | 271 | 0.81 | 0.250 | **296** | 0.09 | 0.025 |
| 0063 | 293 | 14.5% | 268 | **0.18** | **0.025** | 262 | 0.46 | 0.048 | 268 | 0.92 | 0.605 | **288** | 0.32 | 0.100 |
| 0130 | 285 | 14.4% | 199 | 5.12 | 0.618 | 192 | **0.20** | **0.023** | 187 | 1.20 | 0.349 | **281** | 2.00 | 0.909 |
| 0080 | 284 | 12.9% | 162 | **0.58** | 0.109 | 139 | 0.59 | **0.096** | 278 | 2.62 | 0.868 | **283** | 1.92 | 0.237 |
| 0240 | 298 | 11.9% | **295** | 0.64 | 3.479 | 275 | 3.13 | 0.265 | 278 | 1.31 | 0.470 | 294 | **0.39** | **0.135** |
| 0007 | 290 | 11.7% | **283** | 1.53 | 0.150 | 172 | **0.91** | **0.041** | 277 | 1.24 | 0.174 | **290** | **0.19** | **0.035** |

## 4.4 RESULTS

Our results on the MegaDepth and 1DSfM test sets and comparisons to baselines are shown in Tables 1 and 2, respectively. For each scene, we also report the number of input images ($N_c$), the fraction of outlier track points, and compare our VGPA method against the baselines in terms of number of registered images, mean rotation error (in degrees), translation error, and runtime.

Across both benchmarks, VGPA outperforms the deep factorization baseline RESfM on most scenes, achieving lower rotation and translation errors. Compared to classical pipelines, VGPA is competitive with Theia and GLOMAP, and often surpasses them on both metrics. In terms of coverage, VGPA registers a larger fraction of images than RESfM, though typically fewer than GLOMAP.

Table 2: **1DSFM experiment.** For each scene, we show the number of input images (denoted $N_c$) and the fraction of outliers. For each model, we show the number of images used for reconstruction ($N_r$) and mean values of the rotation (in degrees) and translation errors. Winning results are marked in bold and underlined. Yellow represents the best result among the deep-based algorithms and green among the classical algorithms.

| Scene | $N_c$ | Outliers% | **Ours** | | | RESFM | | | Theia | | | GLOMAP | | |
| --- | --- | --- | --- | --- | --- | --- | --- | --- | --- | --- | --- | --- | --- | --- |
| | | | $N_r$ | Rot | Trans | $N_r$ | Rot | Trans | $N_r$ | Rot | Trans | $N_r$ | Rot | Trans |
| Alamo | 573 | 32.6% | 509 | **1.50** | **0.342** | 484 | 3.66 | 0.515 | 553 | 4.42 | 1.433 | **557** | 2.45 | 1.520 |
| Ellis Island | 227 | 25.1% | 214 | **0.27** | **0.077** | 214 | 0.82 | 0.122 | 213 | 5.01 | 1.527 | **219** | 0.58 | 0.155 |
| Madrid Metropolis | 333 | 39.4% | 295 | 1.47 | **0.136** | 244 | 8.42 | 0.827 | - | - | - | **320** | **1.22** | 0.242 |
| Montreal Notre Dame | 448 | 31.7% | 425 | **0.34** | **0.073** | 346 | 2.82 | 0.352 | 422 | 4.47 | 1.285 | **444** | 0.60 | 0.211 |
| NYC Library | 330 | 33.6% | 285 | 1.20 | 0.422 | 224 | 3.96 | 0.429 | 314 | 4.06 | 1.141 | **323** | **0.58** | **0.189** |
| Notre Dame | 549 | 35.6% | 519 | **0.64** | **0.065** | 517 | 1.20 | 0.231 | 534 | 3.70 | 0.828 | **543** | 2.73 | 0.389 |
| Piazza del Popolo | 336 | 33.1% | 315 | 4.42 | 0.710 | 249 | **2.20** | **0.186** | 325 | 3.31 | 1.053 | **331** | **0.80** | 0.188 |
| Tower of London | 467 | 27.0% | 454 | 0.78 | 0.073 | 94 | **0.67** | **0.026** | 448 | 6.61 | 1.189 | **466** | 0.81 | 0.138 |
| Vienna Cathedral | 824 | 31.4% | 753 | 19.28 | 1.285 | 479 | **1.52** | **0.112** | 772 | 12.25 | 1.663 | **822** | 2.00 | 2.414 |
| Yorkminster | 432 | 29.0% | 403 | 1.38 | **0.144** | 331 | 14.54 | 1.468 | 390 | 8.35 | 1.916 | **418** | **0.95** | 0.316 |

Following Khatib et al. (2025), we evaluate VGPA on the smaller Strecha and BlendedMVS benchmarks, which provide ground-truth camera poses. As shown in Table 3, VGPA is consistently more accurate than image-based deep baselines (VGGSfM, MASt3R, and VGGT), which typically do not

scale to the larger datasets considered, and it performs on par with classical pipelines (including Theia, COLMAP, and GLOMAP).

Table 3: **Strecha & BlendedMVS datasets.** For each scene we list the number of input images ($N_c$) and outlier fraction. For each method we report the number of registered images ($N_r$), mean rotation error (deg), translation error, and runtime (s). Best is **bold**, second best is underlined.

| Scene | $N_c$ | Out.% | Ours $N_r$ | Rot | Trans | Time | VGGT $N_r$ | Rot | Trans | Time | MASt3R $N_r$ | Rot | Trans | Time | VGGSfM $N_r$ | Rot | Trans | Time | Theia $N_r$ | Rot | Trans | Time | COLMAP $N_r$ | Rot | Trans | Time | GLOMAP $N_r$ | Rot | Trans | Time |
|---|---|---|---|---|---|---|---|---|---|---|---|---|---|---|---|---|---|---|---|---|---|---|---|---|---|---|---|---|---|---|
| | | | | | | | | | | | | | | **Strecha** | | | | | | | | | | | | | | | | |
| entry-P10 | 10 | 4.8 | 10 | **0.004** | **0.0005** | 10.0 | 10 | 0.079 | 0.033 | 16.5 | 10 | 0.442 | 0.055 | 19 | 10 | 0.165 | 0.056 | 10.3 | 10 | 0.024 | 0.008 | **0.9** | 10 | 0.023 | 0.007 | 36.0 | 10 | 0.187 | 0.026 | 12.5 |
| fountain-P11 | 11 | 1.4 | 11 | **0.012** | **0.0005** | 14.7 | 11 | 0.034 | 0.019 | 12.2 | 11 | 0.160 | 0.026 | 22 | 11 | 0.172 | 0.016 | 15.4 | 11 | 0.027 | 0.002 | **1.5** | 11 | 0.027 | 0.003 | 37.0 | 11 | 0.194 | 0.022 | 38.6 |
| Herz-Jesus-P8 | 8 | 1.8 | 8 | **0.009** | **0.0010** | 7.4 | 8 | 0.032 | 0.011 | 12.7 | 8 | 0.363 | 0.037 | 16 | 8 | 0.206 | 0.042 | 8.7 | 8 | 0.025 | 0.005 | **0.6** | 8 | 0.026 | 0.004 | 22.0 | 8 | 0.091 | 0.015 | 5.0 |
| Herz-Jesus-P25 | 25 | 2.8 | 25 | **0.010** | **0.0003** | 12.5 | 25 | 0.048 | 0.007 | 31.9 | 25 | 0.869 | 0.057 | 81 | 25 | 0.158 | 0.046 | 19.6 | 25 | 0.026 | 0.006 | **2.4** | 25 | 0.028 | 0.006 | 60.0 | 25 | 0.138 | 0.013 | 76.6 |
| | | | | | | | | | | | | | | **BlendedMVS** | | | | | | | | | | | | | | | | |
| scene0 | 75 | 2.0 | 74 | 0.019 | 0.0011 | 136 | 75 | 0.041 | 0.017 | 108 | 75 | 0.501 | 0.191 | 516 | 75 | 0.045 | 0.0106 | 61 | 75 | 0.009 | 0.0017 | 49 | 75 | **0.006** | **0.0005** | 106 | 75 | 0.007 | 0.0016 | 198 |
| scene1 | 51 | 1.4 | 51 | 0.341 | 0.0342 | 38 | 51 | 0.101 | 0.050 | 41 | 51 | 0.919 | 0.173 | 1017 | 51 | 0.098 | 0.0112 | 32 | 51 | 0.029 | 0.0099 | 18 | 51 | **0.007** | **0.0003** | 67 | 51 | 0.024 | 0.0102 | 117 |
| scene2 | 33 | 2.2 | 33 | 0.008 | 0.0004 | 19 | 33 | 0.230 | 0.022 | 52 | 33 | 1.972 | 0.130 | 117 | 33 | 0.227 | 0.0180 | 30 | 33 | 0.045 | 0.0098 | **15** | 33 | **0.003** | **0.0002** | 55 | 33 | 0.025 | 0.0060 | 87 |
| scene3 | 66 | 8.8 | 66 | 0.006 | 0.0065 | 65 | 66 | 0.353 | 0.014 | 276 | 66 | 0.927 | 0.045 | 815 | 66 | 0.372 | 0.0174 | 52 | 66 | 0.019 | 0.0018 | **21** | 66 | **0.004** | **0.0002** | 128 | 66 | 0.008 | 0.0017 | 392 |

**Robustness to view-graph density.** We train VGPA using relative poses obtained from **exhaustive** pairwise matching. At test time, we vary the sparsity of the view graph by using NetVLAD retrieval to connect each image only to its top-$k$ nearest neighbors. As shown in Table 4, VGPA maintains accuracy comparable to the exhaustive graph while using far fewer edges. Its performance changes only slightly across a wide range of $k$, as long as the graph remains sufficiently connected.

**Postprocessing (view re-integration).** Since our pipeline may discard some images during the BA stage, we attempt to re-register these views in postprocessing using a lightweight add-back loop. Unregistered views are ranked by connectivity (e.g., number of 2D–3D matches) with the current point cloud. For each candidate, we estimate its pose from the available 2D–3D correspondences and refine it with a short local BA applied to its neighboring views. The process repeats until no further views can be added. Table 8 in the appendix compares *Ours* and *Ours + post-processing* in terms of $N_r$, mean rotation error (deg), and mean translation error, showing that the add-back step increases the number of registered cameras with minimal runtime overhead (about 1 second per added view).

**Uncalibrated image collections.** Table 5 compares two settings: (i) using ground-truth intrinsics and (ii) starting from an approximate calibration ($f_x, f_y$ proportional to image size, principal point at the image center) and optimizing intrinsics jointly with the extrinsics during bundle adjustment. While self-calibration incurs a small accuracy drop relative to ground-truth intrinsics, VGPA remains competitive and maintains high performance.

**Qualitative results.** Figure 1 shows 3D reconstructions and camera parameters obtained by VGPA for two scenes with more than 1,000 images; in both scenes we register almost all images. These results demonstrate that our method produces superior reconstructions and effectively handles outliers compared to the baselines. Moreover, VGPA is not limited by the number of images, unlike image-based deep methods such as VGGT and VGGSfM. Additional qualitative results are provided in the Appendix.

**Runtime.** Table 6 reports runtimes on the identical point tracks produced by our preprocessing. VGPA is substantially faster than COLMAP, GLOMAP, and Theia, and remains competitive in throughput. Importantly, these gains come without sacrificing reconstruction quality: VGPA

Table 4: **Robustness to graph density.** For each scene we list the number of input images ($N_c$). Our default setting uses *Exhaustive SIFT*, and we also report results with *NetVLAD@K + SIFT* for different values of $K$. For all methods, we show the number of registered images ($N_r$), mean rotation error (deg), and mean translation error.

| Scene | $N_c$ | Ours (Exhaustive SIFT) $N_r$ | Rot | Trans | NetVLAD@20 $N_r$ | Rot | Trans | NetVLAD@30 $N_r$ | Rot | Trans | NetVLAD@40 $N_r$ | Rot | Trans |
|---|---|---|---|---|---|---|---|---|---|---|---|---|---|
| Alamo | 573 | 509 | 1.5 | 0.34 | 533 | 1.29 | 0.77 | 525 | 1.29 | 0.56 | 525 | 1.32 | 1.23 |
| Ellis Island | 227 | 214 | 0.27 | 0.08 | 219 | 0.24 | 0.07 | 218 | 0.266 | 0.08 | 218 | 0.28 | 0.08 |
| Madrid Metropolis | 333 | 295 | 1.47 | 0.14 | 312 | 2.82 | 0.20 | 303 | 2.34 | 0.28 | 309 | 2.39 | 0.11 |

Table 5: **Impact of Camera Intrinsics (Known vs. Estimated).** For each scene, we report the number of input images ($N_c$) and the outlier fraction. We compare our method with known intrinsics vs. without intrinsics (optimized) and report $N_r$, mean rotation error (deg), and mean translation error. Best results are in **bold**.

| Scene | $N_c$ | Out.% | Ours (w/ intrinsics) | | | Ours (w/o intrinsics) | | |
|---|---|---|---|---|---|---|---|---|
| | | | $N_r$ | Rot | Trans | $N_r$ | Rot | Trans |
| *BlendedMVS scenes (shared intrinsics)* | | | | | | | | |
| scene0 | 75 | 2.0 | **74** | **0.019** | **0.0011** | **74** | **0.019** | 0.0019 |
| scene1 | 51 | 1.4 | **51** | 0.341 | **0.0342** | **51** | **0.338** | 0.0400 |
| scene2 | 33 | 2.2 | **33** | **0.008** | **0.0004** | **33** | 0.025 | 0.0049 |
| scene3 | 66 | 8.8 | **66** | **0.006** | 0.0065 | **66** | 0.010 | **0.0014** |
| *MegaDepth scenes (not shared intrinsics)* | | | | | | | | |
| 0012 | 299 | 16.3 | **295** | **0.63** | **0.071** | 293 | 0.70 | 0.235 |
| 0024 | 365 | 23.0 | **313** | 3.38 | 0.772 | 298 | **0.80** | **0.384** |
| 0048 | 486 | 24.3 | **500** | **0.29** | **0.026** | 486 | 0.68 | 0.128 |
| 0083 | 635 | 31.3 | **614** | **0.06** | **0.005** | 601 | 0.56 | 0.219 |

achieves accuracy and coverage comparable to classical pipelines, demonstrating that learned view-graph pose averaging is efficient at scale.

Table 6: **Runtime.** Given the same point tracks, we compare the runtime of our proposed method (VGPA) to RESfM and classical methods, including COLMAP, Theia, and GLOMAP.

| Scene | $N_c$ | Outliers% | Ours | | | RESfM | | | COLMAP | | | Theia | | | GLOMAP | | |
|---|---|---|---|---|---|---|---|---|---|---|---|---|---|---|---|---|---|
| | | | Total (Mins) | $N_r$ | $N_r/t \uparrow$ | Total (Mins) | $N_r$ | $N_r/t \uparrow$ | Total (Mins) | $N_r$ | $N_r/t \uparrow$ | Total (Mins) | $N_r$ | $N_r/t \uparrow$ | Total (Mins) | $N_r$ | $N_r/t \uparrow$ |
| Alamo | 573 | 32.6 | 4.4 | 509 | **116.2** | 17.2 | 484 | 28.2 | 83.7 | 568 | 6.8 | 13.4 | 553 | 41.4 | 40.0 | 557 | 13.9 |
| Ellis Island | 227 | 25.1 | 1.1 | 214 | **194.4** | 2.8 | 214 | 75.9 | 14.9 | 223 | 15.0 | 1.1 | 213 | 193.6 | 7.7 | 219 | 28.6 |
| Madrid Metropolis | 333 | 39.4 | 1.7 | 295 | **172.5** | 5.8 | 244 | 42.1 | 25.1 | 323 | 12.9 | – | – | – | 7.1 | 320 | 45.2 |
| Montreal Notre Dame | 448 | 31.7 | 2.8 | 425 | **151.8** | 6.1 | 346 | 56.7 | 35.9 | 447 | 12.5 | 3.7 | 422 | 114.6 | 13.5 | 444 | 32.9 |
| Notre Dame | 549 | 35.6 | 2.9 | 519 | **179.5** | 22.2 | 517 | 23.3 | 72.6 | 546 | 7.5 | 11.6 | 534 | 46.0 | 21.1 | 543 | 25.8 |
| NYC Library | 330 | 33.6 | 1.3 | 285 | **212.7** | 4.0 | 224 | 55.7 | 26.6 | 330 | 12.4 | 1.5 | 314 | 204.2 | 7.3 | 323 | 44.5 |
| Piazza del Popolo | 336 | 33.1 | 1.1 | 315 | **277.6** | 2.7 | 249 | 92.6 | 9.6 | 334 | 34.9 | 3.0 | 325 | 108.8 | 5.9 | 331 | 56.0 |
| Tower of London | 467 | 27.0 | 3.3 | 454 | 137.6 | 5.9 | 94 | 15.9 | 65.0 | 467 | 7.2 | 3.1 | 448 | **142.5** | 23.5 | 466 | 19.8 |
| Vienna Cathedral | 824 | 31.4 | 7.5 | 753 | **101.0** | 23.9 | 479 | 20.0 | 98.9 | 824 | 8.3 | 11.2 | 772 | 68.8 | 41.6 | 822 | 19.8 |
| Yorkminster | 432 | 29.0 | 2.9 | 403 | **140.5** | 7.7 | 331 | 42.9 | 31.4 | 419 | 13.3 | 2.9 | 390 | 135.3 | 14.8 | 418 | 28.2 |
| *Mean* | – | – | 2.9 | 417 | **168.4** | 9.8 | 318 | 45.3 | 46.4 | 448 | 13.1 | 5.7 | 441 | 117.2 | 18.2 | 444 | 31.5 |

**Ablations.** Ablations confirm that each core component of our method is critical. Removing subset sampling substantially increases both rotation and translation errors, showing its importance for robustness. Excluding DINO appearance cues or reducing the number of GNN layers also leads to a modest decline. Most importantly, fine-tuning yields a large improvement, reducing both rotation and translation errors to their lowest values. See Table 7, where we report errors *before* the final BA refinement.

Table 7: Ablation study reporting mean rotation and translation errors *before* final BA refinement.

| | Mean Rotation Error ($\downarrow$) | Mean Translation Error ($\downarrow$) |
|---|---|---|
| Ours w/o subset sampling | 12.9 | 2.5 |
| Ours w/o image features | 9.8 | 2.2 |
| Ours w/ 2 layers | 10.1 | 2.2 |
| Ours (base model) | 9.5 | 2.1 |
| Proposed (with fine-tuning) | **1.9** | **0.5** |

# 5 CONCLUSION

We present VGPA, an unsupervised deep *pose-averaging* network for multiview SfM. The design includes a permutation-equivariant pose-averaging module that enforces consistency of pairwise rotations and translation directions while incorporating image-level context. Additional 3D point triangulation and robust BA refinement ensure high accuracy and recover the 3D structure. Across challenging benchmarks (including MegaDepth, 1DSfM), VGPA outperforms deep methods and remains competitive with strong classical pipelines while maintaining high camera coverage. It is also *fast*: on the same point tracks, VGPA is substantially faster than COLMAP and GLOMAP, and modestly faster than Theia, while scaling to large image collections. A lightweight view re-integration sweep reintroduces part of the few remaining discarded views with negligible overhead.

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

APPENDIX

# A  QUALITATIVE RESULTS

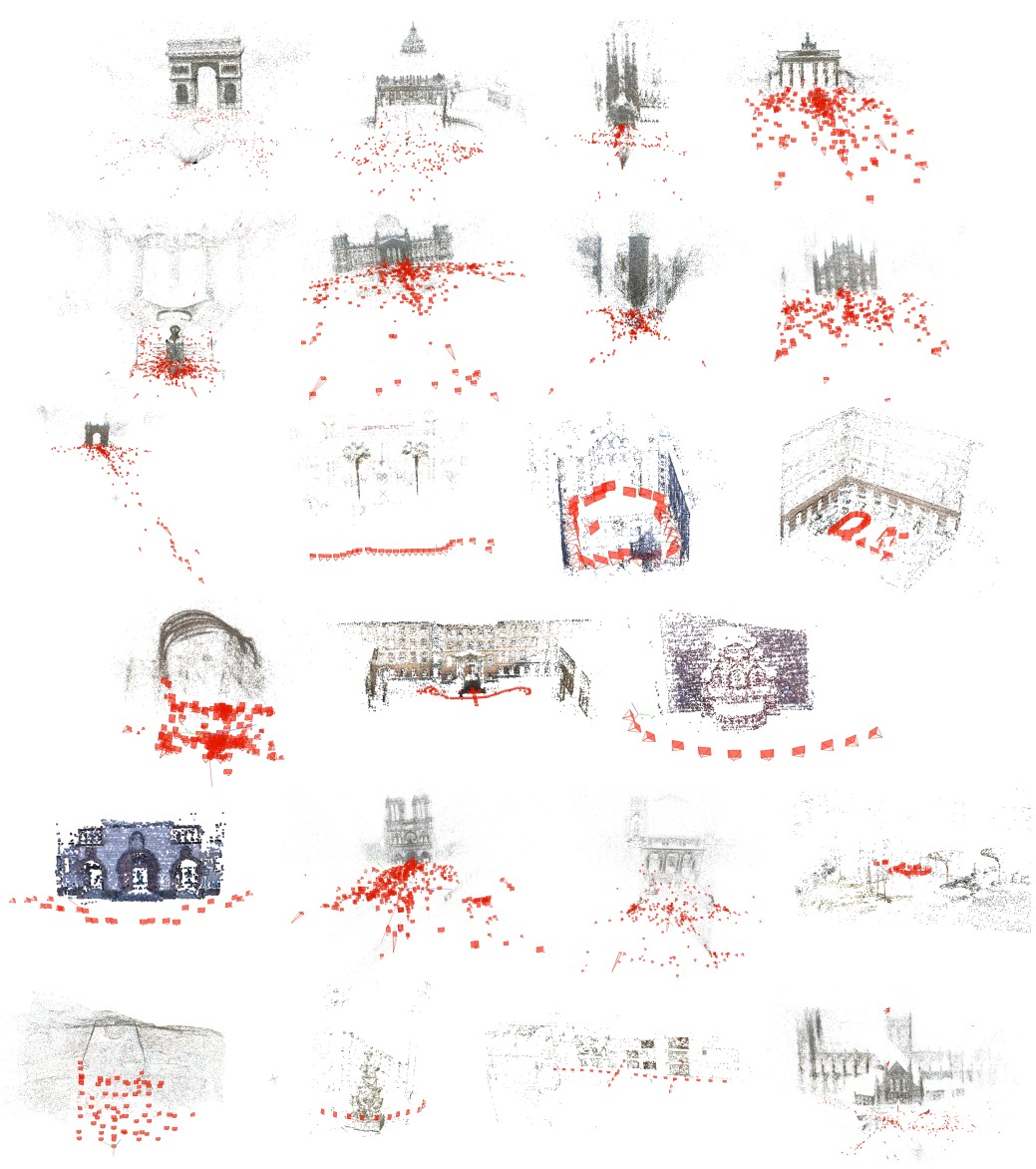

Figure 3: Example reconstructions from the proposed VGPA on various datasets.

## Use of Large Language Models (LLMs)

We used a large language model (ChatGPT) solely for language polishing, i.e., improving grammar, clarity, and style of sentences.

## B   Implementation details

**Code and data.** Our code and preprocessed data will be made publicly available.

**Framework.** We train and evaluate on NVIDIA A100 GPUs (80 GB). The implementation uses PyTorch (Paszke et al., 2019) and the Adam optimizer (Kingma & Ba, 2014) with gradient normalization.

**Training.** Each epoch iterates over all training scenes. For every scene, we uniformly sample (without replacement) 10%–20% of the images to form the training subgraph. A held-out validation set is used for early stopping. Validation and test evaluations use the complete view graph. Training on MegaDepth takes approximately 8 hours on a single A100. We fix the random seed to 20.

**Architecture details.** The encoder uses 3 edge-conditioned message passing layers with 256 channels (nodes and edges) and ReLU activations. The camera head $H_{\mathrm{cams}}$ is a 3-layer MLP with 256 channels.

**Hyperparameter search.** We sweep over (1) learning rate $\{10^{-2}, 10^{-3}, 10^{-4}\}$, (2) network width $\{128, 256, 512\}$ for the encoder and heads, and (3) number of layers $\{2, 3, 4, 5\}$.

**Bundle adjustment.** We use Ceres Solver Agarwal et al. with a Huber loss (scale 0.1) for robustness, following Khatib et al. (2025). In each BA round, we cap the number of iterations at 300 or stop earlier on convergence.

## C   Constructing point tracks

We follow the preprocessing in Khatib et al. (2025) to construct point tracks; see their Appendix for full details.

## D   Performance of other Deep-Based Methods on the 1DSfM Dataset

As shown in the table below, all three 3D geometric foundation models perform poorly on the 1DSfM dataset, in contrast to our method.

Table 8: **Deep-based methods on the 1DSfM dataset.** For each scene we list the number of input images ($N_c$). For each deep model (TTT3R, CUT3R, FAST3R) we report the mean rotation error (degrees) and mean translation error. Best results are bold and underlined.

| Scene | $N_c$ | TTT3R Rot | TTT3R Trans | CUT3R Rot | CUT3R Trans | FAST3R Rot | FAST3R Trans |
|---|---|---|---|---|---|---|---|
| Alamo | 573 | **16.08** | **3.650** | 22.32 | 4.266 | 40.37 | 3.990 |
| Ellis Island | 227 | **8.85** | **2.333** | 12.53 | 3.171 | 11.74 | 3.201 |
| Madrid Metropolis | 333 | **14.83** | **2.258** | 18.81 | 3.189 | 67.37 | 3.574 |
| Montreal Notre Dame | 448 | 13.25 | **1.230** | 16.12 | 2.134 | **10.79** | 2.052 |
| NYC Library | 330 | **7.67** | **1.656** | 10.26 | 1.912 | 11.90 | 2.408 |
| Notre Dame | 549 | **11.88** | **1.430** | 15.33 | 1.694 | 16.44 | 2.241 |
| Piazza del Popolo | 336 | 23.13 | **2.063** | **22.63** | 2.092 | 32.48 | 2.568 |
| Tower of London | 467 | **29.69** | 3.647 | 29.85 | **3.607** | 59.53 | 3.658 |
| Vienna Cathedral | 824 | 43.76 | 2.978 | 41.58 | 2.802 | **29.49** | **2.561** |
| Yorkminster | 432 | **16.45** | **2.223** | 23.00 | 2.992 | 20.26 | 2.463 |

# E  ADDITIONAL RESULTS

Here we present the view re-integration results (Table 9). Tables 11 and 10 report the AUC (Area Under the recall Curve) scores—computed from the maximum of the relative rotation and translation errors between every image pair—across different thresholds (in degrees), for both the MegaDepth and 1DSfM experiments. Tables 12 and 13 report the corresponding median errors for the two datasets.

Table 9: **MegaDepth: Effect of View Re-Integration.** We report the number of input images ($N_c$), outlier fraction, registered images ($N_r$), and mean rotation and translation errors for our method (*Ours*) and with the add-back step (*Ours + Post-Processing*).

| Scene | $N_c$ | Outliers% | $N_r$ | Ours Rot | Trans | $N_r$ | Ours + post-processing Rot | Trans |
|---|---|---|---|---|---|---|---|---|
| 0238 | 522 | 44.6% | 488 | 4.50 | 0.686 | 511 | 4.43 | 0.681 |
| 0060 | 528 | 41.6% | 518 | 0.07 | 0.014 | 526 | 0.08 | 0.016 |
| 0197 | 870 | 40.7% | 641 | 1.28 | 0.271 | 749 | 1.33 | 0.291 |
| 0094 | 763 | 40.1% | 663 | 0.66 | 0.101 | 708 | 1.24 | 0.134 |
| 0265 | 571 | 38.8% | 345 | 2.93 | 0.998 | 476 | 3.50 | 1.077 |
| 0083 | 635 | 31.3% | 614 | 0.06 | 0.005 | 628 | 0.07 | 0.007 |
| 0076 | 558 | 30.5% | 543 | 0.09 | 0.016 | 553 | 0.11 | 0.018 |
| 0185 | 368 | 30.0% | 358 | 0.10 | 0.022 | 364 | 0.11 | 0.022 |
| 0048 | 512 | 24.2% | 500 | 0.29 | 0.026 | 507 | 0.29 | 0.026 |
| 0024 | 356 | 23.0% | 313 | 3.38 | 0.772 | 342 | 3.39 | 0.781 |
| 0223 | 214 | 17.0% | 208 | 2.75 | 0.195 | 213 | 3.56 | 0.289 |
| 5016 | 28 | 16.9% | 28 | 0.08 | 0.015 | 28 | 0.08 | 0.015 |
| 0046 | 440 | 14.6% | 439 | 0.54 | 0.071 | 440 | 0.54 | 0.071 |
| 1001 | 285 | 43.9% | 265 | 1.89 | 3.840 | 274 | 1.86 | 3.938 |
| 0231 | 296 | 42.2% | 261 | 0.24 | 0.030 | 271 | 0.45 | 0.061 |
| 0411 | 299 | 29.9% | 270 | 0.12 | 0.018 | 289 | 0.13 | 0.021 |
| 0377 | 295 | 27.5% | 232 | 0.30 | 0.035 | 253 | 0.32 | 0.036 |
| 0102 | 299 | 25.8% | 297 | 0.18 | 0.031 | 299 | 0.18 | 0.031 |
| 0148 | 287 | 24.6% | 211 | 0.93 | 0.037 | 225 | 2.28 | 0.229 |
| 0147 | 298 | 24.6% | 282 | 1.99 | 0.153 | 292 | 1.98 | 0.153 |
| 0446 | 298 | 22.1% | 292 | 0.22 | 0.019 | 297 | 0.24 | 0.021 |
| 0022 | 297 | 21.2% | 277 | 0.29 | 0.044 | 287 | 0.29 | 0.044 |
| 0327 | 298 | 21.0% | 291 | 0.12 | 0.014 | 293 | 0.12 | 0.014 |
| 0015 | 284 | 20.6% | 243 | 0.52 | 0.058 | 255 | 0.68 | 0.111 |
| 0455 | 298 | 19.8% | 290 | 0.39 | 0.078 | 298 | 0.52 | 0.092 |
| 0496 | 297 | 19.2% | 279 | 0.37 | 0.033 | 290 | 0.38 | 0.035 |
| 1589 | 299 | 17.4% | 296 | 0.11 | 0.010 | 298 | 0.11 | 0.010 |
| 0012 | 299 | 16.3% | 295 | 0.63 | 0.071 | 298 | 1.04 | 0.122 |
| 0019 | 299 | 15.4% | 291 | 0.37 | 0.020 | 297 | 0.52 | 0.026 |
| 0063 | 293 | 14.5% | 268 | 0.18 | 0.025 | 274 | 0.20 | 0.026 |
| 0130 | 285 | 14.4% | 199 | 5.12 | 0.618 | 207 | 4.97 | 0.622 |
| 0080 | 284 | 12.9% | 162 | 0.58 | 0.109 | 164 | 0.61 | 0.126 |
| 0240 | 298 | 11.9% | 295 | 0.64 | 3.479 | 297 | 0.64 | 3.485 |
| 0007 | 290 | 11.7% | 283 | 1.53 | 0.150 | 287 | 1.51 | 0.148 |

Table 10: **1DSfM experiment (AUC).** For each scene, we list the number of input images ($N_c$) and the fraction of outliers. For each model, we report the AUC values at different error thresholds (in degrees). Winning results are marked in **bold and underlined**.

| Scene | $N_c$ | Outliers% | Ours | | | | | Theia | | | | | GLOMAP | | | | |
|---|---|---|---|---|---|---|---|---|---|---|---|---|---|---|---|---|---|
| | | | @1 | @3 | @5 | @10 | @30 | @1 | @3 | @5 | @10 | @30 | @1 | @3 | @5 | @10 | @30 |
| Alamo | 573 | 32.6% | **0.444** | **0.623** | **0.676** | **0.730** | 0.804 | 0.002 | 0.037 | 0.093 | 0.228 | 0.499 | 0.092 | 0.346 | 0.482 | 0.647 | **0.833** |
| Ellis Island | 227 | 25.1% | **0.399** | **0.739** | **0.832** | **0.908** | **0.962** | 0.000 | 0.006 | 0.023 | 0.120 | 0.440 | 0.071 | 0.371 | 0.539 | 0.729 | 0.901 |
| Madrid Metropolis | 333 | 39.4% | **0.564** | **0.731** | **0.788** | **0.845** | **0.914** | 0.014 | 0.135 | 0.241 | 0.406 | 0.650 | 0.163 | 0.502 | 0.626 | 0.750 | 0.876 |
| Montreal Notre Dame | 448 | 31.7% | **0.532** | **0.788** | **0.853** | **0.908** | **0.952** | 0.001 | 0.028 | 0.093 | 0.263 | 0.553 | 0.090 | 0.390 | 0.549 | 0.724 | 0.890 |
| NYC Library | 330 | 33.6% | **0.577** | **0.777** | **0.839** | **0.899** | **0.954** | 0.006 | 0.080 | 0.161 | 0.307 | 0.566 | 0.142 | 0.494 | 0.634 | 0.778 | 0.910 |
| Notre Dame | 549 | 35.6% | **0.425** | **0.684** | **0.782** | **0.872** | **0.950** | 0.015 | 0.158 | 0.293 | 0.487 | 0.726 | 0.101 | 0.419 | 0.566 | 0.719 | 0.864 |
| Piazza del Popolo | 336 | 33.1% | **0.422** | **0.553** | 0.598 | 0.643 | 0.699 | 0.025 | 0.140 | 0.226 | 0.368 | 0.608 | 0.203 | 0.524 | **0.648** | **0.775** | **0.899** |
| Tower of London | 467 | 27.0% | **0.437** | **0.675** | **0.757** | **0.833** | **0.901** | 0.002 | 0.039 | 0.093 | 0.209 | 0.474 | 0.114 | 0.453 | 0.600 | 0.750 | 0.897 |
| Vienna Cathedral | 824 | 31.4% | **0.291** | **0.414** | 0.449 | 0.483 | 0.521 | 0.000 | 0.001 | 0.008 | 0.049 | 0.269 | 0.053 | 0.346 | **0.499** | **0.664** | **0.846** |
| Yorkminster | 432 | 29.0% | **0.507** | **0.763** | **0.834** | **0.899** | **0.954** | 0.000 | 0.011 | 0.038 | 0.115 | 0.357 | 0.104 | 0.427 | 0.600 | 0.765 | 0.905 |
| Mean | 451 | 31.9% | **0.460** | **0.675** | **0.741** | **0.802** | 0.861 | 0.007 | 0.064 | 0.127 | 0.255 | 0.514 | 0.113 | 0.427 | 0.574 | 0.730 | **0.882** |

Table 11: **MegaDepth experiment (AUC).** For each scene, we show the number of input images ($N_c$) and the fraction of outliers. For each model, we report the AUC values at different error thresholds (in degrees). Winning results are marked in **bold and underlined**.

| Scene | $N_c$ | Outliers% | Ours | | | | | Theia | | | | | GLOMAP | | | | |
|---|---|---|---|---|---|---|---|---|---|---|---|---|---|---|---|---|---|
| | | | @1 | @3 | @5 | @10 | @30 | @1 | @3 | @5 | @10 | @30 | @1 | @3 | @5 | @10 | @30 |
| 0238 | 522 | 44.6% | **0.322** | 0.449 | 0.503 | 0.559 | 0.748 | 0.063 | 0.338 | 0.495 | 0.679 | 0.863 | 0.298 | **0.552** | **0.653** | **0.761** | **0.884** |
| 0060 | 528 | 41.6% | **0.807** | **0.904** | **0.932** | **0.959** | **0.982** | 0.300 | 0.592 | 0.702 | 0.811 | 0.912 | 0.676 | 0.847 | 0.893 | 0.933 | 0.969 |
| 0197 | 870 | 40.7% | 0.297 | 0.411 | 0.524 | 0.697 | 0.881 | 0.086 | 0.338 | 0.505 | 0.693 | 0.872 | **0.555** | **0.752** | **0.817** | **0.879** | **0.940** |
| 0094 | 763 | 40.1% | **0.708** | **0.851** | **0.890** | **0.926** | **0.959** | 0.325 | 0.610 | 0.708 | 0.807 | 0.904 | 0.468 | 0.696 | 0.772 | 0.846 | 0.921 |
| 0265 | 571 | 38.8% | **0.000** | **0.005** | **0.015** | **0.063** | **0.309** | 0.000 | 0.001 | | 0.037 | 0.267 | **0.000** | 0.000 | 0.001 | 0.009 | 0.165 |
| 0083 | 635 | 31.3% | **0.885** | **0.954** | **0.969** | **0.981** | **0.992** | 0.504 | 0.748 | 0.817 | 0.883 | 0.948 | 0.765 | 0.901 | 0.935 | 0.964 | 0.987 |
| 0076 | 558 | 30.5% | **0.747** | **0.879** | **0.915** | **0.950** | **0.980** | 0.133 | 0.450 | 0.599 | 0.754 | 0.897 | 0.510 | 0.753 | 0.830 | 0.902 | 0.963 |
| 0185 | 368 | 30.0% | **0.821** | **0.910** | **0.930** | **0.951** | **0.973** | 0.285 | 0.627 | 0.742 | 0.846 | 0.937 | 0.641 | 0.839 | 0.889 | 0.933 | 0.970 |
| 0048 | 512 | 24.2% | **0.843** | **0.934** | **0.955** | **0.974** | **0.988** | 0.397 | 0.690 | 0.785 | 0.873 | 0.949 | 0.698 | 0.864 | 0.907 | 0.945 | 0.975 |
| 0024 | 356 | 23.0% | **0.505** | **0.687** | **0.740** | 0.785 | 0.821 | 0.153 | 0.424 | 0.555 | 0.709 | 0.873 | 0.362 | 0.619 | 0.719 | **0.827** | **0.931** |
| 0223 | 214 | 17.0% | **0.592** | **0.781** | **0.836** | **0.886** | **0.926** | 0.014 | 0.182 | 0.342 | 0.551 | 0.783 | 0.330 | 0.601 | 0.703 | 0.807 | 0.908 |
| 5016 | 28 | 16.9% | **0.790** | **0.896** | **0.928** | **0.959** | **0.984** | 0.413 | 0.707 | 0.793 | 0.876 | 0.952 | 0.508 | 0.770 | 0.835 | 0.899 | 0.959 |
| 0046 | 440 | 14.6% | **0.934** | **0.971** | **0.979** | 0.985 | 0.989 | 0.530 | 0.793 | 0.861 | 0.918 | 0.965 | 0.896 | 0.962 | 0.977 | **0.988** | **0.996** |
| 0099 | 299 | 47.4% | 0.256 | 0.530 | 0.638 | 0.757 | 0.878 | 0.011 | 0.082 | 0.172 | 0.348 | 0.619 | **0.526** | **0.707** | **0.769** | **0.835** | **0.909** |
| 1001 | 285 | 43.9% | **0.046** | **0.232** | **0.347** | **0.495** | **0.690** | 0.000 | 0.001 | | 0.005 | 0.051 | 0.000 | 0.001 | 0.003 | 0.020 | 0.132 |
| 0231 | 296 | 42.2% | **0.608** | **0.822** | **0.883** | **0.934** | **0.975** | 0.063 | 0.304 | 0.467 | 0.655 | 0.842 | 0.417 | 0.678 | 0.762 | 0.843 | 0.921 |
| 0411 | 299 | 29.9% | **0.699** | **0.870** | **0.914** | **0.949** | **0.979** | 0.188 | 0.469 | 0.600 | 0.753 | 0.902 | 0.379 | 0.633 | 0.727 | 0.828 | 0.928 |
| 0377 | 295 | 27.5% | **0.770** | **0.887** | **0.916** | **0.940** | **0.961** | 0.198 | 0.471 | 0.596 | 0.736 | 0.883 | 0.567 | 0.754 | 0.824 | 0.889 | 0.941 |
| 0102 | 299 | 25.8% | **0.774** | **0.897** | **0.931** | **0.961** | **0.985** | 0.169 | 0.384 | 0.474 | 0.596 | 0.785 | 0.547 | 0.735 | 0.805 | 0.876 | 0.946 |
| 0147 | 298 | 24.6% | **0.731** | **0.844** | **0.873** | **0.903** | **0.935** | 0.055 | 0.324 | 0.468 | 0.618 | 0.771 | 0.000 | 0.000 | 0.000 | 0.000 | 0.000 |
| 0148 | 287 | 24.6% | **0.681** | **0.819** | **0.859** | **0.903** | **0.948** | 0.049 | 0.199 | 0.280 | 0.376 | 0.499 | 0.296 | 0.422 | 0.472 | 0.529 | 0.591 |
| 0446 | 298 | 22.1% | **0.671** | **0.845** | **0.892** | **0.936** | **0.972** | 0.053 | 0.303 | 0.460 | 0.646 | 0.840 | 0.465 | 0.717 | 0.799 | 0.876 | 0.947 |
| 0022 | 297 | 21.2% | **0.704** | **0.878** | **0.921** | **0.958** | **0.986** | 0.194 | 0.502 | 0.631 | 0.767 | 0.900 | 0.473 | 0.717 | 0.797 | 0.875 | 0.949 |
| 0327 | 298 | 21.0% | **0.589** | **0.686** | **0.710** | **0.761** | 0.872 | 0.040 | 0.359 | 0.538 | 0.722 | **0.884** | 0.474 | 0.631 | 0.683 | 0.733 | 0.776 |
| 0015 | 284 | 20.6% | **0.780** | **0.888** | **0.917** | **0.942** | **0.965** | 0.134 | 0.359 | 0.478 | 0.617 | 0.784 | 0.572 | 0.775 | 0.839 | 0.899 | 0.950 |
| 0455 | 298 | 19.8% | **0.739** | **0.864** | **0.897** | **0.925** | 0.955 | 0.209 | 0.513 | 0.644 | 0.779 | 0.905 | 0.584 | 0.798 | 0.862 | 0.920 | **0.966** |
| 0496 | 297 | 19.2% | **0.739** | **0.879** | **0.916** | **0.950** | **0.977** | 0.080 | 0.348 | 0.498 | 0.671 | 0.849 | 0.441 | 0.697 | 0.786 | 0.871 | 0.942 |
| 1589 | 299 | 17.4% | **0.670** | **0.860** | **0.912** | **0.952** | **0.980** | 0.157 | 0.385 | 0.489 | 0.621 | 0.797 | 0.583 | 0.746 | 0.812 | 0.885 | 0.951 |
| 0012 | 299 | 16.3% | **0.810** | **0.900** | **0.922** | **0.943** | 0.961 | 0.087 | 0.359 | 0.499 | 0.663 | 0.840 | 0.645 | 0.834 | 0.887 | 0.934 | **0.971** |
| 0104 | 284 | 16.2% | **0.575** | **0.667** | **0.693** | **0.716** | **0.735** | 0.127 | 0.328 | 0.430 | 0.538 | 0.642 | 0.445 | 0.592 | 0.634 | 0.674 | 0.717 |
| 0019 | 299 | 15.4% | **0.817** | **0.901** | **0.924** | 0.948 | 0.969 | 0.243 | 0.536 | 0.649 | 0.764 | 0.888 | 0.740 | 0.884 | **0.924** | **0.960** | **0.986** |
| 0063 | 293 | 14.5% | **0.692** | **0.814** | **0.846** | 0.877 | 0.949 | 0.106 | 0.376 | 0.526 | 0.704 | 0.879 | 0.460 | 0.724 | 0.805 | **0.885** | **0.956** |
| 0130 | 285 | 14.4% | **0.631** | **0.744** | **0.776** | **0.802** | 0.830 | 0.057 | 0.282 | 0.424 | 0.604 | 0.828 | 0.347 | 0.479 | 0.549 | 0.666 | **0.835** |
| 0080 | 284 | 12.9% | **0.753** | **0.890** | **0.923** | **0.952** | **0.980** | 0.034 | 0.154 | 0.278 | 0.448 | 0.720 | 0.259 | 0.384 | 0.512 | 0.733 | 0.905 |
| 0240 | 298 | 11.9% | **0.683** | **0.843** | **0.890** | **0.934** | **0.972** | 0.141 | 0.405 | 0.536 | 0.698 | 0.867 | 0.368 | 0.613 | 0.709 | 0.815 | 0.923 |
| 0007 | 290 | 11.7% | **0.803** | **0.893** | **0.917** | **0.938** | 0.963 | 0.069 | 0.415 | 0.591 | 0.761 | 0.902 | 0.659 | 0.829 | 0.881 | 0.931 | **0.974** |
| Mean | 364 | 25.4% | **0.652** | **0.780** | **0.820** | **0.863** | **0.915** | 0.157 | 0.399 | 0.518 | 0.654 | 0.806 | 0.471 | 0.653 | 0.716 | 0.783 | 0.852 |

Table 12: **MegaDepth experiment.** For each scene, we show the number of input images (denoted $N_c$) and the fraction of outliers. For each model, we show the number of images used for reconstruction (denoted $N_r$) and **median** values of the rotation (in degrees) and translation errors. (Above the middle rule are Group 1 scenes with $< 1000$ images; below are Group 2 scenes with $> 1000$ images, subsampled to 300 for testing.) Winning results are marked in bold and underlined. Yellow represents the best result among the deep-based algorithms and green among the classical algorithms.

| Scene | $N_c$ | Outliers% | Ours | | | RESfM | | | Theia | | | GLOMAP | | |
|---|---|---|---|---|---|---|---|---|---|---|---|---|---|---|
| | | | $N_r$ | Rot | Trans | $N_r$ | Rot | Trans | $N_r$ | Rot | Trans | $N_r$ | Rot | Trans |
| 0238 | 522 | 44.6% | 488 | 1.62 | 0.123 | 283 | 0.72 | 0.043 | 506 | 0.54 | 0.109 | 499 | 0.22 | 0.043 |
| 0060 | 528 | 41.6% | 518 | 0.02 | 0.004 | 503 | 0.14 | 0.011 | 525 | 0.26 | 0.039 | 522 | 0.04 | 0.012 |
| 0197 | 870 | 40.7% | 641 | 0.96 | 0.125 | 667 | 2.06 | 0.133 | 855 | 0.77 | 0.118 | 814 | 0.13 | 0.016 |
| 0094 | 763 | 40.1% | 663 | 0.26 | 0.019 | 537 | 0.38 | 0.015 | 742 | 0.21 | 0.033 | 717 | 0.20 | 1.957 |
| 0265 | 571 | 38.8% | 345 | 1.75 | 0.445 | 346 | 0.74 | 0.209 | 554 | 4.11 | 1.651 | 558 | 6.66 | 1.889 |
| 0083 | 635 | 31.3% | 614 | 0.03 | 0.002 | 596 | 0.15 | 0.009 | 632 | 0.15 | 0.013 | 614 | 0.04 | 0.007 |
| 0076 | 558 | 30.5% | 543 | 0.04 | 0.005 | 524 | 0.11 | 0.010 | 549 | 0.44 | 0.058 | 541 | 0.08 | 0.017 |
| 0185 | 368 | 30.0% | 358 | 0.04 | 0.004 | 350 | 0.04 | 0.006 | 365 | 0.31 | 0.037 | 365 | 0.11 | 0.012 |
| 0048 | 512 | 24.2% | 500 | 0.11 | 0.005 | 474 | 2.16 | 0.098 | 507 | 0.21 | 0.020 | 506 | 0.06 | 0.007 |
| 0024 | 356 | 23.0% | 313 | 1.57 | 0.087 | 309 | 0.58 | 0.046 | 355 | 0.24 | 0.091 | 339 | 0.07 | 0.045 |
| 0223 | 214 | 17.0% | 208 | 1.07 | 0.047 | 204 | 1.56 | 0.078 | 212 | 0.89 | 0.152 | 214 | 0.41 | 0.046 |
| 5016 | 28 | 16.9% | 28 | 0.04 | 0.003 | 28 | 0.10 | 0.005 | 28 | 0.07 | 0.019 | 28 | 0.04 | 0.016 |
| 0046 | 440 | 14.6% | 439 | 0.05 | 0.002 | 399 | 0.78 | 0.028 | 434 | 0.16 | 0.016 | 440 | 0.02 | 0.002 |
| 1001 | 285 | 43.9% | 265 | 0.66 | 2.698 | 251 | 1.41 | 0.276 | 276 | 4.85 | 2.893 | 281 | 3.29 | 2.645 |
| 0231 | 296 | 42.2% | 261 | 0.07 | 0.007 | 246 | 0.38 | 0.014 | 286 | 0.58 | 0.072 | 279 | 0.20 | 0.021 |
| 0411 | 299 | 29.9% | 270 | 0.07 | 0.009 | 273 | 0.07 | 0.009 | 293 | 0.19 | 0.079 | 269 | 0.09 | 0.036 |
| 0377 | 295 | 27.5% | 232 | 0.09 | 0.005 | 210 | 0.28 | 0.014 | 269 | 0.29 | 0.075 | 268 | 0.23 | 0.021 |
| 0102 | 299 | 25.8% | 297 | 0.06 | 0.006 | 284 | 0.07 | 0.007 | 294 | 1.03 | 0.114 | 293 | 0.04 | 0.013 |
| 0147 | 298 | 24.6% | 282 | 0.80 | 0.030 | 207 | 2.07 | 0.088 | 284 | 1.10 | 0.064 | 290 | 1.78 | 2.056 |
| 0148 | 287 | 24.6% | 211 | 0.43 | 0.018 | 197 | 0.54 | 0.024 | 275 | 3.01 | 0.301 | 283 | 3.09 | 1.301 |
| 0446 | 298 | 22.1% | 292 | 0.10 | 0.005 | 288 | 0.41 | 0.013 | 289 | 0.61 | 0.073 | 296 | 0.14 | 0.020 |
| 0022 | 297 | 21.2% | 277 | 0.12 | 0.009 | 274 | 0.13 | 0.011 | 296 | 0.28 | 0.065 | 281 | 0.08 | 0.023 |
| 0327 | 298 | 21.0% | 291 | 0.05 | 0.004 | 271 | 0.11 | 0.006 | 288 | 0.73 | 0.087 | 290 | 7.14 | 0.333 |
| 0015 | 284 | 20.6% | 243 | 0.15 | 0.009 | 215 | 0.27 | 0.021 | 244 | 0.42 | 0.084 | 274 | 0.11 | 0.014 |
| 0455 | 298 | 19.8% | 290 | 0.11 | 0.007 | 293 | 0.18 | 0.010 | 294 | 0.36 | 0.047 | 298 | 0.14 | 0.017 |
| 0496 | 297 | 19.2% | 279 | 0.15 | 0.007 | 281 | 0.13 | 0.006 | 285 | 0.61 | 0.080 | 291 | 0.16 | 0.028 |
| 1589 | 299 | 17.4% | 296 | 0.03 | 0.002 | 290 | 0.08 | 0.003 | 288 | 0.32 | 0.057 | 299 | 0.03 | 0.007 |
| 0012 | 299 | 16.3% | 295 | 0.10 | 0.006 | 287 | 0.39 | 0.023 | 129 | 0.56 | 0.092 | 295 | 0.20 | 0.017 |
| 0019 | 299 | 15.4% | 291 | 0.17 | 0.007 | 250 | 0.04 | 0.004 | 271 | 0.31 | 0.030 | 296 | 0.04 | 0.004 |
| 0063 | 293 | 14.5% | 268 | 0.05 | 0.004 | 262 | 0.26 | 0.013 | 268 | 0.45 | 0.063 | 288 | 0.17 | 0.017 |
| 0130 | 285 | 14.4% | 199 | 2.71 | 0.089 | 192 | 0.10 | 0.005 | 187 | 0.63 | 0.072 | 281 | 0.94 | 0.535 |
| 0080 | 284 | 12.9% | 162 | 0.25 | 0.009 | 139 | 0.27 | 0.010 | 278 | 1.84 | 0.335 | 283 | 1.71 | 0.169 |
| 0240 | 298 | 11.9% | 295 | 0.10 | 3.371 | 275 | 1.56 | 0.090 | 278 | 0.47 | 0.057 | 294 | 0.17 | 0.041 |
| 0007 | 290 | 11.7% | 283 | 0.40 | 0.022 | 172 | 0.23 | 0.010 | 277 | 0.69 | 0.071 | 290 | 0.06 | 0.006 |

Table 13: **1DSFM experiment.** For each scene, we show the number of input images (denoted $N_c$) and the fraction of outliers. For each model, we show the number of images used for reconstruction ($N_r$) and **median** values of the rotation (in degrees) and translation errors. Winning results are marked in bold and underlined. Yellow represents the best result among the deep-based algorithms and green among the classical algorithms.

| Scene | $N_c$ | Outliers% | Ours | | | RESfM | | | Theia | | | GLOMAP | | |
|---|---|---|---|---|---|---|---|---|---|---|---|---|---|---|
| | | | $N_r$ | Rot | Trans | $N_r$ | Rot | Trans | $N_r$ | Rot | Trans | $N_r$ | Rot | Trans |
| Alamo | 573 | 32.6% | 509 | 0.42 | 0.018 | 484 | 0.97 | 0.037 | 553 | 2.29 | 0.539 | 557 | 0.61 | 0.144 |
| Ellis Island | 227 | 25.1% | 214 | 0.16 | 0.033 | 214 | 0.32 | 0.036 | 213 | 3.85 | 0.712 | 219 | 0.46 | 0.087 |
| Madrid Metropolis | 333 | 39.4% | 295 | 0.27 | 0.016 | 244 | 4.42 | 0.193 | - | - | - | 320 | 0.53 | 0.096 |
| Montreal Notre Dame | 448 | 31.7% | 425 | 0.16 | 0.020 | 346 | 1.00 | 0.056 | 422 | 2.63 | 0.808 | 444 | 0.40 | 0.158 |
| NYC Library | 330 | 33.6% | 285 | 0.58 | 0.038 | 224 | 1.48 | 0.074 | 314 | 1.65 | 0.360 | 323 | 0.46 | 0.075 |
| Notre Dame | 549 | 35.6% | 519 | 0.29 | 0.012 | 517 | 0.55 | 0.025 | 534 | 1.54 | 0.133 | 543 | 1.15 | 0.130 |
| Piazza del Popolo | 336 | 33.1% | 315 | 2.11 | 0.120 | 249 | 0.80 | 0.034 | 325 | 1.15 | 0.342 | 331 | 0.28 | 0.084 |
| Tower of London | 467 | 27.0% | 454 | 0.23 | 0.011 | 94 | 0.48 | 0.012 | 448 | 3.23 | 0.527 | 466 | 0.42 | 0.071 |
| Vienna Cathedral | 824 | 31.4% | 753 | 11.78 | 0.527 | 479 | 0.48 | 0.016 | 772 | 9.32 | 0.838 | 822 | 0.61 | 0.206 |
| Yorkminster | 432 | 29.0% | 403 | 0.62 | 0.022 | 331 | 4.67 | 0.299 | 390 | 4.26 | 0.948 | 418 | 0.60 | 0.069 |

