# OpenReview forum: "VGPA: Deep View-Graph Pose Averaging for Structure-from-Motion"
_ICLR.cc/2026/Conference — Submitted to ICLR 2026_

### Official Review · Reviewer_pRcq · 2025-10-28

**Soundness:** 3
**Presentation:** 3
**Contribution:** 2
**Rating:** 6
**Confidence:** 3

**Summary:**

This paper addresses the classical problem of Structure from Motion (SfM). Specifically, it proposes a pose-averaging structure-from-motion method consisting of three main components.

1. The first component is preprocessing. Given a set of input images, the preprocessing stage includes feature extraction using a DINOv2 backbone, followed by view graph construction and relative pose estimation. These steps prepare the data for the following stages.
2. The second component involves graph neural network–based pose averaging. In this stage, pose averaging is performed within the latent space of a graph neural network. A pose regression head then predicts the actual poses from the latent representations.
3. Finally, a bundle adjustment step refines the pose-averaged results, producing the final 3D reconstruction output of the Structure from Motion pipeline.

Experiments show that this method is both faster and more accurate than several state-of-the-art baselines. The experiments were conducted on multiple well-known datasets, and the results appear convincing.

In summary, the key contribution of this paper lies in the graph neural network–based pose averaging module, which introduces a novel approach to improving accuracy and efficiency in SfM. Overall, I find the experiments convincing, although I have a few questions listed in the following sections. My overall recommendation for this paper is a weak accept.

**Strengths:**

- The paper is well written and easy to follow.
- It presents a sound and well-motivated approach to addressing the challenging problem of pose averaging in structure-from-motion (SfM).
- The proposed graph neural network for pose averaging looks novel to me.

**Weaknesses:**

1. My main concern lies in the **scalability** of the proposed method. Since it relies on a graph neural network to process multiple nodes, I would like to see results on larger-scale datasets—for example, those containing 10,000+ images. The current evaluation primarily focuses on datasets with 1,000 images or fewer, and for scenes exceeding this number, the data are subsampled to 1,000 images. I believe that evaluating the method on a truly large-scale dataset would better demonstrate its advantages, especially given that the proposed pose-averaging approach is designed to handle well such scenarios.

**Questions:**

1. Could you evaluate the proposed method on large-scale scenes containing over 10,000 images, and compare the results with relevant baselines?
2. For baselines such as VGGT, Mast3R, and VGGSfM, which do not rely on precomputed graph information or coarse relative poses, the proposed method appears to have an inherent advantage by leveraging these inputs. How is this difference in input requirements addressed to ensure a fair comparison?

---

> ### Author Response · Authors · 2025-12-03
>
> We thank the reviewer for his thoughtful feedback.
>
>
> **W1 + Q1. Scalability to Large Scenes**
>
> Compared to existing deep methods, VGPA already handles significantly larger scenes. We searched MegaDepth for bigger collections and applied VGPA to scene 0141, which contains 2,296 images. Our method successfully processed this scene in about 20.1 minutes, whereas COLMAP and GLOMAP require much more time. The resulting reconstruction is highly accurate, with a **rotation error of 0.18° and a translation error of 0.02**, and the method registered **2,264 out of 2,296 images**. We obtained similar results for scene 0036, which contains 2,605 images.
>
> Importantly, GPU memory usage scales favorably, so the model can, in principle, handle datasets with 10,000+ images. Although it is difficult to find public datasets of this size for evaluation, we generated synthetic inputs corresponding to 10,000 images and did not encounter any out-of-memory issues.
>
> **Q2. Using Relative Poses**
>
> Please note that Mast3R and VGGSfM do use relative poses. For example, Mast3R first estimates pairwise relative poses as part of its pipeline, and VGGSfM similarly computes relative poses internally during its reconstruction process. In addition, please note that all these methods, including VGGT, were trained on much larger and more diverse datasets compared to the 30 scenes used to train VGPA. Therefore, in this case, VGGT also has an inherent advantage due to large-scale training, whereas our method uses only automatically computed (and noisy) relative poses derived directly from the input images.

---

### Official Review · Reviewer_H4CN · 2025-10-31

**Soundness:** 2
**Presentation:** 2
**Contribution:** 2
**Rating:** 2
**Confidence:** 4

**Summary:**

The paper proposes to integrate a global edge-conditioned, permutation-equivariant GNN into the  Structure-from-Motion (SfM) pipeline  that takes as input noisy pairwise relative poses and outputs globally consistent camera extrinsics. The supervision of the model comes from enforcing consistency between predicted global poses and input relative poses, making the model fine-tunable at inference time.  After initial pose prediction, the poses are refined with a robust bundle adjustment initialized with the camera poses predicted by the network and the triangulated points.

To demonstrate the efficiency, robustness and scalability of the method the authors provide experiments on MegaDepth, 1DSfM, Strecha, and BlendedMVS showing competitive results with classical pipelines.

**Strengths:**

The permutation-equivariant, edge-conditioned GNN architecture that processes noisy pairwise relative poses to output globally consistent camera extrinsics trained in an unsupervised via relative-pose consistency seems novel and interesting.

The model is tested on several datasets and the different components are ablated in order to estimate their effect.

**Weaknesses:**

Although the pose regression network does not use point trajectories, predict 3D points, or rely on reprojection loss, all results are provided with the final BA algorithm, which does use them. Table 7 is the only table that presents the results obtained directly with the proposed network. Therefore, it is difficult to assess the efficiency of the GNN itself.

The model appears significantly more complex than recent feedforward methods like Dust3R and its successors such as VGGT. Actually, it seems more complex than even the classical SfM pipeline. It still requires keypoint extraction, feature matching, RANSAC, and pairwise essential matrix computation to build the GNN graph. A DINOv2 feature extractor is also added to inject image-level context to the nodes. After the GNN’s feed-forward pass, it requires fine-tuning on the target scene. Finally, to obtain the final results  a robust bundle adjustment is performed, initialized with the network’s predicted camera poses and triangulated points. Given all these additional steps, it may seem surprising that this model is faster than classical SfM pipeline.  It's unclear if the authors considered all these steps in the runtime computation during their claims about speed.

The model does not seems to generalizing well between datasets as the real gain seems to come from the final fine-tuning (Table 7). Note that it is not précised on which dataset the ablations were conducted and in Table 7 the results obtained with the final BA is missing so it is  unclear how much the added components really improves the pipeline.

 The tables are too complex, containing numerous numbers that can make it challenging to assess the performance at a glance. An additional line displaying average ranks would be beneficial for providing a comprehensive perspective on the results.
It is also important to note that evaluating rotation and translation errors in isolation can be misleading, as the interdependence between these errors (e.g., a small rotation error paired with a large translation error, or vice versa) may obscure the true performance of the method.  A combined metric like Mean Average Accuracy (mAA)@30, such as that used in the DUSt3R paper (section 4.2), or metrics used for visual localization, such as the percentages of camera poses correctly estimated within a pair of thresholds for rotation and translation (available at https://www.visuallocalization.net), would be beneficial to provide a more accurate comparison between methods.

The proposed model, unlike newer Geometric Foundation Models (GFM) such as Dust3R and VGGT, relies on explicitly provided camera intrinsic parameters, which are not typically required in feed-forward methods designed for uncalibrated real-world image collections. While the authors argue that self-calibration in VGPA incurs only a minor accuracy loss compared to ground-truth intrinsics, this claim is validated only on the BlendedMVS dataset (Table 5), where intrinsics are consistent within scenes making their estimation easy.  However, the model’s ability to handle real-world uncalibrated datasets—where each image has unique, unknown intrinsics—remains untested.

**Questions:**

The authors state that one of the advantages of their method compared to GFMs lies in the scalability. However, on the one hand, their preprocessing step and the size of their GNN graph increase quadratically with the number of images. On the other hand, several GFM methods, such as Fast3R, CUT3R, and MUST3R, have been shown  to be able to process scenes with hundres or even thousands of images.

Generally speaking, since GFM methods sometimes exhibit shortcomings in the accurate estimation of poses, it would have been more interesting to investigate in the paper whether a GNN, such as the one proposed in this article, could be trained  through self-learning based on the consistency of the relative pose to improve the poses estimated by GFM  models without using any preprocessing or BA postprocessing.

Most GFM models uses CO3Dv2 and RealEstate10K datasets to evaluate pose estimation. It would be therefore important to run the proposed method on these datasets and compare it with the same metrics. In particular to show also the results obtained before the BA.

Finally, as the VGPA pipeline integrates the last step BA, it would also be relevant to evaluate the accuracy of the 3D reconstruction, again preferably on the datasets used by the GFM papers such as DTU, 7Scenes, TUM-RGBD, NRGBD ...

---

> ### Author Response · Authors · 2025-12-03
>
> We thank the reviewer for his thoughtful feedback.
>
> **W1. GNN Evaluation**
>
> The GNN operates on relative poses that are themselves derived from point trajectories, so it indirectly benefits from trajectory information. Importantly, almost all SfM pipelines report results after bundle adjustment, because BA is necessary to reach final geometric accuracy. Table 7 isolates the GNN’s behavior, but the main evaluation follows standard SfM practice by reporting post-BA performance.
>
> **W2. Fair Runtime Measurement**
>
> Please note that our preprocessing steps, such as computing point tracks and pairwise relative poses, are identical to those required by classical SfM systems, including COLMAP, GLOMAP, and Theia. Therefore, following standard practice, the reported runtime does not include preprocessing time for **any** of the methods.
> Similarly, Mast3R also performs expensive preprocessing: it runs the network on image pairs, which is substantially slower than running SIFT + RANSAC as in our pipeline. VGGT, on the other hand, relies on CoTracker to obtain the tracks, and its preprocessing time is comparable to ours. Thus, the runtime reported for all methods is measured fairly and consistently.
>
> **W3. Generalization Concerns**
>
> Fine-tuning is indeed helpful, especially for large scenes, but this does not imply that the GNN fails to generalize. To show this, we compare the errors obtained using (i) our pretrained GNN and (ii) a randomly initialized GNN. The pretrained model performs substantially better, demonstrating strong cross-dataset generalization.
> Regarding Table 7, we intentionally did not apply BA because it would reduce the differences between ablations and obscure the specific contribution of each GNN component. The goal of this table is to isolate the behavior of the GNN itself, not the full pipeline. In general, better BA initialization also leads to faster BA convergence.
>
>
>
> **W4. Additional Metrics**
>
> Thanks for this comment. Adding additional metrics can indeed be helpful. We follow the common practice of reporting mean rotation and translation errors; however, we agree that newer combined metrics provide a more comprehensive view. We have therefore added two new tables (Tables 10 and 11) in the appendix, reporting AUC@k (k = 1, 3, 5, 30) on MegaDepth and 1DSfM. This metric, which measures the percentage of relative poses whose maximum rotation and translation errors fall below a threshold, complements the original tables and provides a clearer overall comparison. **The two tables show that our method outperforms GLOMAP and Theia on almost all scenes.**
>
> **W5. Handling Uncalibrated / Non-Shared Intrinsics**
>
> We thank the reviewer for bringing this point to our attention. We have updated Table 5 with additional experiments on randomly selected MegaDepth scenes, where each image has different and often unknown intrinsics, reflecting real-world uncalibrated photo collections. As shown in the new table, our method maintains similar performance under this setting, supporting the claim that VGPA can handle non-shared, uncalibrated intrinsics in practice.
>
>
> **Q1. Scalability Compared to GFMs**
>
> Thanks for raising this point. We have added results for CUT3R, Fast3R, and the concurrent TTT3R on the 1DSfM dataset. As shown, all three methods struggle in large-scale scenes; they generally perform best on video-like sequences rather than Internet photo collections with high viewpoint variation.
> Regarding scalability, the GNN is efficient, and its graph does not need to be fully quadratic. When using NetVLAD to select relevant neighbors, the graph becomes sparse and grows approximately linearly with the number of images. Even in the quadratic case, the growth is in the number of images, rather than in the number of per-image tokens as in many GFM architectures, which keeps memory and compute manageable at larger scales.
>
> **Q2. Potential Extensions to GFM Methods**
>
> We agree that this is an interesting direction. Generally, the more accurate the relative poses are, the better the overall results become. This is a promising future extension, but it is outside the scope of the current work.
>
> **Q3. Reconstruction Quality**
>
> We note that our method, as well as the classical SfM baselines, produces sparse reconstructions, so direct comparison to dense reconstructions is not standard. Instead, SfM methods are typically evaluated using camera accuracy and sparse reprojection error. In our case, the reprojection error after BA is below one pixel, indicating that the resulting 3D structure is accurate.

---

> ### Author Response · Authors · 2025-12-03
> **W3. Generalization Concerns**
>
> **Comparison: Pretrained GNN vs. Randomly Initialized GNN**
> (Using the same four MegaDepth scenes from Table 5)
>
> We report the mean rotation errors (without applying BA):
>
> | Scene ID | #Images | **Pretrained GNN** Rot (°) | **Random Init GNN** Rot (°) |
> |----------|---------|-----------------------------|-------------------------------|
> | 0083     | 635     | **11.0**                    | 29.9                      |
> | 0012     | 299     | **9.9**                    | 27.1                        |
> | 0024     | 365     | **6.2**                    | 17.6                        |
> | 0048     | 486     | **11.1**                    | 30.3                        |
>
> This shows that our GNN exhibits generalization ability.

---

### Official Review · Reviewer_Z29H · 2025-10-31

**Soundness:** 3
**Presentation:** 3
**Contribution:** 3
**Rating:** 6
**Confidence:** 3

**Summary:**

This paper proposes VGPA, a permutation-equivariant GNN-based network for view-graph pose averaging. VGPA takes as input pairwise relative poses and image features and predicts globally consistent camera extrinsics in a feedforward manner. It can be trained without ground-truth camera pose labels, relying solely on relative-pose consistency.

Extensive experiments demonstrate the effectiveness of the proposed method. VGPA outperforms prior deep-based approaches while maintaining high efficiency, and it matches or even surpasses competitive optimization-based methods in accuracy.

**Strengths:**

1. The proposed method is simple and straightforward. It can be trained without ground-truth camera labels, highlighting its potential for scalability.
2. The experiments are comprehensive, showing the proposed method’s strong performance in both accuracy and efficiency compared to previous deep and non-deep baselines. The ablation studies further validate the contribution of each component.
3. The writing is clear and easy to follow.

**Weaknesses:**

1. Table 3 briefly shows results under sparse image settings (e.g., fewer than 20 views). The reviewer is curious about the method’s robustness in even more challenging scenarios, where pairwise relative camera poses may become highly noisy due to insufficient correspondences (e.g., in object-centric CO3D settings). Would incorporating image features improve robustness in such cases?
2. It would be interesting to examine the scaling behavior of the proposed method. For instance, comparing models trained with larger versus smaller datasets could demonstrate whether the method exhibits desirable scaling properties.

**Questions:**

1. In L372–375, the authors mention that the method is robust across a wide range of k values. Are the numbers of nearest neighbors (k) reported in Table 4?
2. In Table 7, Row 1 shows that subset sampling is quite important for the proposed method. Do the authors have any intuition for this behavior? Additionally, could they conduct an ablation study on the effect of different proportions of kept samples on performance?

---

> ### Author Response · Authors · 2025-12-03
>
> We thank the reviewer for his thoughtful feedback.
>
> **W1. Sparse CO3D-Style Scenarios**
>
> The extremely sparse CO3D-style setting was introduced mainly for deep methods that specifically target this scenario, since classical SfM struggles to obtain enough correspondences with SIFT when fewer than ~20 views are available. In such cases, our method would face similar limitations because the relative poses would be too noisy. As with COLMAP, using more robust keypoint detectors for texture-less scenes would improve correspondence quality, and we expect our method to benefit similarly. However, this setting is outside the scope of our work, which focuses on a deep-based approach that performs reliably on large-scale, diverse scenes while remaining competitive with COLMAP.
>
> **W2.**
>
> That is an interesting direction. We experimented with adding a few more scenes to the training set, but the improvement was small. We believe that meaningful scaling gains would require training on much larger and more diverse datasets, similar to recent GFM (3D Geometric Foundation Models) models that use a mixture of 30+ datasets and require extensive GPU resources and weeks of training. Exploring this properly is beyond our current computational budget, so we leave it for future work.
>
> **Q1. Using NetVLAD with Different k Values**
>
> We updated the table (5) to include results for k = 20, 30, and 40. In practice, k = 30 is commonly used for Internet photo collections such as 1DSfM. Across all these values, our method achieves performance similar to that of exhaustive pairwise matching, which demonstrates its robustness to the choice of k.
>
>
> **Q2. Subset Sampling**
>
> We believe that subset sampling acts as a strong data augmentation mechanism. Since the model is trained on a relatively small number of scenes, sampling random subgraphs exposes it to many more graph configurations, thereby improving its robustness and generalization. We experimented with different proportions of kept samples and observed similar performance, likely because all settings still expose the network to a large variety of random configurations.

---

### Official Review · Reviewer_z73w · 2025-11-01

**Soundness:** 2
**Presentation:** 3
**Contribution:** 2
**Rating:** 2
**Confidence:** 4

**Summary:**

This paper presents a deep-learning based global Structure-from-Motion framework based that use learned view-graph aggregation, using a permutation-equivariant, edge-conditioned graph neural network. The GNN starts with the standard noisy pose-graph and outputs globally consistent camera extrinsics. The GNN is trained on a relative-pose consistency objective in a self-supervised manner. A view re-integration step is described to incrase camera coverage by examining discarded images. Results are shown on multiple datasets.

**Strengths:**

The GNN framework/design seems better than prior GNN-based methods.

**Weaknesses:**

Results seem very mixed. VGPA does well only in half the cases presented, even over GNN-based methods. More importantly, the design choices are not explained or motivated well, making it all appear ad-hoc.

**Questions:**

Overall, I am not enthusiastic about the work as presented in this paper. Here are some of my comments and observations. These aren't specific questions to be answered by the authors, but points that I have problems with.
- No clear motivation or intuition given on the GNN design. I would expect more reasons given to convince the ICLR audience of the method. $\phi_e$, $\phi_m$, and $\psi$ are just presented. Why these designs and why not some other ones? Why was $\log R_{ij}$ used?
- What is the structure of $\phi_e$, $\phi_m$, and $\psi$? Are the fully connected? Linear? Activation function?
- The pose-regression head $H_{cams}$ is described as a 3-layer MLP. What is the architecture of the others?
- The loss (Eq 1) treats the RANSAC-estimates from COLMAP of R and t as the ground-truth and the loss aims for estimates to match those. I am not sure it is correct to describe it as "unsupervised"; ground-truth is estimated using COLMAP. This also ensures the presented method cannot be better than RANSAC method.
- My major dissatisfaction is about the results. VGPA seems better only in half the cases in Table 1. In particular, classical methods register more cameras.  How do we make a case for VGPA based on these results? Are all “test”scenes of MegaDepth reported here? Does an average of all make sense?
- The results on MegaDepth scenes is varied too, with some doing better, others worse. Is there anything to be learned by carefully examining the scenes that do well and otherwise? Any observations on those?
- No clear advantage of VGPA shown other datasets.  Strecha results seem better, but not BMVS. How can we say it does better? Any deeper analysis?
- Based on highly mixed results, what is a strong case for VGPA? I am not able to find a clear case.
- Re-integration is a tangential point. Other methods also may be able to do it. What is unique about it for VGPA?
- Similarly, what is the importance of "uncalibrated" results (Tab 5) there?
- I find the presentation of "$N_r/t$" strange and even misleading. It makes it appear as if the method can register more cameras if run for more time. Why is it relevant?
- VGPA seems to do well in total time for most datasets. I would recommend reporting on that fact alone.
- Regarding ablations, I think the choices to use $\phi_e$, $\phi_m$, $\psi$, and $H$ at least as relevant. What is the impact of other design choices for them?

---

> ### Author Response · Authors · 2025-12-03
>
> We thank the reviewer for his thoughtful feedback.
>
> **Clarification:**
>
> It is important to note that none of the baselines we compare against are GNN-based methods. VGPA is, to our knowledge, the only GNN-based approach that directly tackles the full SfM problem, apart from GASfM. We do not include GASfM as a baseline because, as shown in RESfM on the same datasets we use, it fails when the input point tracks contain outliers.
>
> **W1. Mixed Results**
>
> Regarding the claim that the results are mixed, we would like to clarify the following.
>  On BlendedMVS and Strecha—datasets with ground-truth poses—VGPA and COLMAP achieve the same level of accuracy, and the very small numerical differences are not significant at that scale. On the large-scale 1DSfM and MegaDepth datasets (hundreds to thousands of images), our method achieves accuracy comparable to state-of-the-art classical SfM systems while being substantially faster: for example, 13× faster than COLMAP on 1DSfM and 5× faster than GLOMAP, and still faster than the “fast” Theia pipeline while outperforming Theia in accuracy.
>
> VGPA achieves accuracy comparable to top classical methods on large-scale datasets while being much faster. Unlike many recent deep methods such as VGGT, it can run on these large scenes. Even models that do operate at this scale (e.g., CUT3R, TTT3R, Fast3R) often struggle on such scenes and yield poor accuracy, as shown in the newly added Table 8 in the appendix.
>
>
>
> **Q1a. Motivation for the use of GNN**
>
> The motivation for using a GNN is that it naturally aligns with both the structure and symmetries of the SfM problem. Our inputs are pairwise relative poses, which form a view-graph where images are nodes and relative geometries are edges. A GNN is therefore the most appropriate architecture, since it operates directly on graphs and is permutation-equivariant, which aligns with the fact that image order carries no meaning.
>
> This choice is also consistent with classical pose-averaging methods, which explicitly formulate the problem on a view-pose graph. Using a GNN is thus a natural deep-learning extension of this classical formulation.
> Regarding the design of the individual GNN components, we follow standard and common choices, MLP-based edge and node update functions with residual connections, which we found to be the most stable and effective in practice.
>
>
> **Q1b. GNN Design Details**
>
> The number of layers was chosen through hyperparameter search. All modules (`φ_e`, `φ_m`, `ψ`) use 3-layer MLPs with ReLU activations. These architectural details are provided in the appendix.
>
>
>
> **Q2. Relative poses from RANSAC**
>
> We would like to clarify that our loss does not use relative poses from COLMAP’s full reconstruction. Instead, we use only the pair-wise relative poses obtained from SIFT matching followed by a RANSAC, which is the same preprocessing step used by COLMAP, GLOMAP, Theia, and other classical SfM pipelines. These pairwise relative poses are not globally optimized and often contain significant noise, inconsistencies across loops, and incorrect translations. Our method learns to denoise, reconcile, and globally regularize these noisy pairwise estimates.
> Therefore, the supervision is more accurately described as self-supervised using noisy proxy labels, not using COLMAP’s optimized outputs.
>
>
> **Q3. Observations from the Results and Number of Registered Images**
>
> Even in cases where another method, such as GLOMAP, achieves slightly lower errors on a specific scene, VGPA still produces highly accurate reconstructions.
> Regarding the number of registered images: VGPA discards more images than GLOMAP (though far fewer than RESfM). As shown in the appendix, a large fraction of these discarded images can be re-integrated in a post-processing step, further reducing this gap.
>
> **Q4. Reintegration Post-processing**
>
> We agree. Reintegration is a technical addition rather than a core contribution, which is why we place it in the appendix and do not include it in the main paper’s results. Other methods could also apply this, and we do not claim uniqueness in this regard.
>
>
> **Q5. Unclibrated setting**
>
> We agree that the uncalibrated setting is less common today since intrinsics are often available from EXIF. However, they are occasionally missing or unreliable, so we include these results simply to show that the method can still operate in such cases. This is not a core focus of the paper.
>
>
> **Q6. time/$N_r$**
>
>
> The metric represents time divided by the number of registered cameras, following the RESfM evaluation protocol. Its purpose is to normalize runtime across methods that register different numbers of cameras, not to suggest that longer runtime leads to more registrations.

---

### Author Response · Authors · 2025-12-03

We deeply appreciate the reviewers for their insightful feedback!

Dear AC, unfortunately, due to the technical issue that occurred on OpenReview and the subsequent closure of the discussion option, we did not have the opportunity to engage with the reviewers. However, we did our best to address all of their concerns thoroughly.

**We would like to emphasize the importance of our contribution:**

**Significance:** Our work introduces an effective deep network-based approach for global SfM that achieves accuracy comparable to state-of-the-art classical methods while being

**Significantly faster**, especially on large-scale scenes. At the same time, recent Geometric Foundation Models (GFMs) struggle in this setting, making VGPA a practical and efficient alternative for large and challenging image collections. We believe this marks an important step in advancing deep learning as a leading technique for large-scale SfM.

**Novelty:** Our GNN-based method is the first to directly address the full pose-averaging problem (rotation and translation) in a deep-learning framework. We hope this contribution will serve as a foundation for further progress in deep-based global SfM.

**We have made the following revisions in the text:**

1. We updated Table 5 with several MegaDepth scenes to demonstrate that our method can handle non-shared and unknown intrinsics.


2. We added an evaluation of three recent 3D geometric foundation models (CUT3R, TTT3R, and Fast3R) on the 1DSfM dataset in Table 8, showing that they perform very poorly in this setting.


3. At the request of the reviewers, we added two new tables (Tables 10 and 11) in the appendix reporting AUC metrics at different thresholds for our experiments on 1DSfM and MegaDepth. **These results show that our method outperforms GLOMAP and Theia on almost all scenes.**

---

### Meta-Review · Area_Chair_tjDd · 2025-12-23

**Summary:**

The paper proposes VGPA, a GNN-based global SfM pipeline for pose averaging on a view-graph built from noisy pairwise relative poses, with optional fine-tuning and a final bundle adjustment (BA). Reviewers’ concerns that drive reject recommendation concentrate on three themes:

Multiple reviewers feel key design choices in the GNN modules and training objective are insufficiently motivated, making the approach appear ad-hoc rather than principled for an ICLR audience.

The strongest concerns argue that many headline results are reported after additional steps (fine-tuning + BA), making it hard to isolate the contribution and efficiency of the proposed GNN itself, and to understand whether improvements come from the core model or from downstream optimization.

Reviewers question whether runtime comparisons consistently account for all pipeline components, and whether the results provide a clear, consistent advantage across scenes/datasets. The “registered cameras” gap, reliance on reintegration, and variability across benchmarks make the main claim less compelling.

**Reviewer Concerns:**

Concerns addressed by the rebuttal:

Authors clarified MLP depth/activations and moved specifics to the appendix.

Authors clarified they use pairwise RANSAC relative poses as noisy proxy targets rather than COLMAP’s globally optimized reconstruction, reframing as self-/proxy-supervision.

Authors added AUC-style metrics at multiple thresholds, improving interpretability.

Authors expanded Table 5 with additional MegaDepth scenes to argue robustness to unknown/non-shared intrinsics.

Authors added experiments on ~2.3k–2.6k image scenes and reported synthetic memory stress tests.

Concerns still outstanding:

Even with added details, the rebuttal does not provide a compelling why-this-design narrative.

The rebuttal argues “comparable accuracy + faster” mainly on large-scale sets, but concerns remain about inconsistency across scenes and the trade-off with coverage/registered cameras. Reintegration is acknowledged as not unique and placed as post-processing, which further blurs the core claim.

Even with Table 7 isolating the network, the paper’s main tables largely reflect the full pipeline. It remains unclear how much of the final performance and speed claims are due to the GNN itself, versus optimization and pipeline engineering.

Authors state preprocessing is excluded for all methods and claim fairness, but the concern persists that end-to-end practicality is hard to assess when multiple expensive steps are involved, and different baselines have different internal costs.

The added ~2–3k image evidence helps, but does not fully address the request for 10k+ real scenes, nor does it conclusively resolve quadratic graph-growth concerns in practice.

Additional appendix tables for GFMs help, but concerns remain about apples-to-apples comparison given differing inputs/assumptions and where the method sits relative to feed-forward foundations vs classical SfM.

**Reviewer Scores:**

Reviewer z73w (score 2, reject) → likely stays 2

Reviewer H4CN (score 2, reject) → likely stays 2

Reviewer Z29H (score 6, weak accept) → likely stays 6

Reviewer pRcq (score 6, weak accept) → likely stays 6

---

### Decision · Program_Chairs · 2026-01-26

Reject